# Directional Changes in the Intestinal Bacterial Community in Black Soldier Fly (*Hermetia illucens*) Larvae

**DOI:** 10.3390/ani11123475

**Published:** 2021-12-06

**Authors:** Xinfu Li, Shen Zhou, Jing Zhang, Zhihao Zhou, Qiang Xiong

**Affiliations:** 1College of Food Science and Light Industry, Nanjing Tech University, Nanjing 211800, China; lixinfu316@126.com (X.L.); zhou_shen0629@126.com (S.Z.); zhouzhihao91@126.com (Z.Z.); 2College of Biotechnology and Pharmaceutical Engineering, Nanjing Tech University, Nanjing 211800, China; zhangjing3737@126.com

**Keywords:** black soldier fly, 16S rRNA sequencing, *Lysinibacillus sphaericus*

## Abstract

**Simple Summary:**

The black soldier fly (BSF), *Hermetia illucens* (Diptera: Stratiomyidae), is renowned for its bioconversion of organic waste into a sustainable source of animal feed. Gut microbes play an essential role in aiding their host during the digestion of complex substrates by possessing metabolic properties that the insect lacks. Microbes that survive the gut passage are candidates for microbes that contribute more to larval development, besides just being a nutrient source. Insect larvae cohabit in some form of symbiosis with microbes. Here, a preliminary experiment was performed to explore the dynamics of the *H. illucens* gut microbiota and the changes in the composition of the bacterial community in organic waste with six different functional strains of the larval feed during rearing. The results showed that the increase in the abundance of *Lysinibacillus* in the experimental group that was exposed to *Lysinibacillus sphaericus* was significantly different to the other groups (*p* < 0.05). The results indicate that *H. illucens* larvae have a stable gut microbiome that does not change significantly during larval development, whereas bacterial communities in the feed residue with the addition of certain bacteria can be slightly affected by rearing.

**Abstract:**

Black soldier fly (BSF) larvae, *Hermetia illucens* (Diptera: Stratiomyidae) have emerged as an efficient system for the bioconversion of organic waste. Intestinal microorganisms are involved in several insect functions, including the development, nutrition, and physiology of the host. In order to transform the intestinal bacterial community of BSF directionally, six different potential functional strains (*Lysinibacillus sphaericus*, *Proteus mirabilis*, *Citrobacter freundii*, *Pseudocitrobacter faecalis*, *Pseudocitrobacter anthropi*, and *Enterococcus faecalis*) were added to aseptic food waste, and aseptic food waste was used without inoculants as a blank control to evaluate the changes in the intestinal microbiota of BSF under artificial intervention conditions. These six strains (which were isolated from the larval intestinal tract in selective media and then identified and screened) may be considered responsible for the functional characteristics of larvae. The results imply that the increase in the abundance of *Lysinibacillus* in the experimental group that was exposed to *Lysinibacillus sphaericus* was significantly different to the other groups (*p* < 0.05). The results revealed that it is feasible to transform the intestinal microbiota of BSF directionally; there are differences in the proliferation of different strains in the intestine of BSF.

## 1. Introduction

The study of insect gut microbes is not only conducive to the development and utilization of insect resources, but it is also beneficial to obtaining bacterial resources with specific functions from the environment of the insect gut. Previous studies have confirmed that the insect gut (of various species) can be used as an effective source for separating important enzymes in industry, such as proteases and other productive enzyme strains [1,2,3,4]. There has been much research on the intestinal microbes of insects, including that of silkworms [5,6], termites [7], and long-horned beetles [8].

In the study and utilization of insects, there are many reports of insects that combine microorganisms to degrade various organic wastes. Relatively few reports focus on changes in the structure of microbial intestinal microorganisms to promote waste degradation. Qi et al. [9] used *Trichoderma viride*, *Saccharomyces cerevisiae*, and *Musca domestica* to transform crop straws, evaluated their impact on housefly rearing performance, and optimized their utilization. Their results showed that the use of *T. viride* and *S. cerevisiae* to ferment crop straw can enhance the biotransformation of crop straw and improve the rearing capacity of housefly larvae.

The microbes in the fly larvae gut have multiple functions that are important to larval development [10]. The functions of gut microbiota impact the development, pathogen resistance, nutrition, and physiology of the host. So far, there is little in-depth understanding of the unique intestinal biotransformation system of insects, particularly regarding the functions of the various symbiotic microorganisms in the intestine. The relationships between insects and symbiotic microorganisms, and the potential science and application values have been rationalized; however, regarding the research into mammalian gut bacteria [11,12,13], there remains much room for development concerning the compounds from insect gut microbes. Enzymes are the most important driver of diet decomposition. There are few reports on the screening of high-efficiency enzyme-producing strains from insect intestinal microbes and the degradation of various organic wastes in combination with insects [10,14,15]. Our focus, with respect to substrate composition, was on the organic compound, whereas, because bacterial communities in the feed residue are affected by the addition of certain bacteria, cellulase, protease, and lipase-secreting bacteria were selected and isolated from the BSF larvae using selection media.

Black soldier fly (BSF) larvae, an important environmental insect, are saprophytic and have a wide range of food sources. BSF larvae can quickly convert organic waste (such as food waste [16,17], poultry manure [18,19,20], straw [21,22], sewage sludge [23], and organic leachates [24,25]) into stable biological fertilizers and their biomass. Additionally, BSF larvae were found to produce broad-spectrum antimicrobial peptides with different antibacterial activities according to the diet [26], which can reduce a variety of pathogens [27]. Changes in diet can also cause BSF larvae to form different bacterial communities in their intestinal tracts [28,29]. BSF larvae can improve the efficiency of conversion when the feed substrate is inoculated with single strains or mixtures of bacteria [30]. For example, inoculating *Lysinibacillus boronitolerans*, *Kocuria marina*, or *Proteus mirabilis* into chicken manure produced larger larvae and reduced manure residue [31]. Fly larvae are symbiotic with microorganisms in a certain form [10]. In addition to being a source of nutrition, microorganisms that survive in the intestinal tract may also greatly contribute to the development of larvae. However, few studies have evaluated their ability to colonize, or have explored the effects of isolated functional bacterial strains on intestinal microbiota.

The intestinal symbiotic bacteria of BSF larvae contribute significantly to the degradation of organic waste [32]. However, our understanding of the intestinal bacteria basic symbiotic capability of BSFL remains fundamental. In the present study, we hypothesized that the *H. illucens* larvae have a relatively stable gut microbiome, whereas bacterial communities can be affected by rearing the feeding substrates with the isolated functional strains from BSFL gut microbiota. Therefore, the objective of this study is to ascertain the early colonization and microbial communities of insect gut bacteria, as affected by some isolated strains, based on 16S rRNA gene surveys.

## 2. Materials and Methods

### 2.1. Source of BSF and Food Waste

The BSF eggs, 10-day-old BSF larvae, and the food waste were provided by Younong Environmental Protection Industry Technology Co., Ltd. (Taizhou, Jiangsu Province, China). The food waste was sterilized at 121 °C for 15 min and then cooled to room temperature for subsequent testing.

### 2.2. Isolation and Screening of Enzyme-Producing Strains

From each batch (of three), guts from 10 randomly selected (of approximately 1000) 10-day-old BSF larvae, were used for bacterial isolation. All isolation procedures were implemented in aerobic conditions. The surface of the 10-day-old BSF larvae was disinfected with 75% alcohol, washed with sterile water several times, and then surgically dissected. At least 10 larvae guts were sampled and homogenized. We took 0.1 g of the intestinal contents and performed a gradient dilution treatment from 10^−3^ to 10^−8^. Then, the dilution plate method and the streak plate method were used to isolate the strains. Carboxymethyl cellulose (CMC)-agar medium (CMC 0.2%, peptone 0.5%, beef extract 0.5% NaCl 0.5%, agar 2%, pH 7.0) was used for screening the cellulolytic bacteria. Milk protein medium (skimmed milk powder 1.5%, peptone 1%, beef extract 3%, NaCl 0.5%, agar 2%, pH 7.0) was used for screening the proteolytic bacteria. Neutral red oil medium (olive oil polyvinyl alcohol emulsion 12%, peptone 1%, beef extract 0.5%, NaCl 0.5%, MgSO_4_ 0.05%, 1.6% neutral red 0.1%, agar 2%, pH 7.0) was used for screening the lipolytic bacteria. These strains were cultured separately, inoculated into 300 mL LB medium in a 500 mL Erlenmeyer flask, and cultured for 48 h at 37 °C. The cultured LB media (with bacteria) were centrifuged at 10,000× *g* to isolate the bacterial cells. The cells collected after centrifugation were suspended in PBS and diluted to the concentration of 10^8^ CFU/mL in distilled water (as described previously by Yu et al. [32]) for further use in the experiments with BSF larvae. Finally, the isolation and identification mainly focused on aerobic bacteria, because anaerobic bacteria are not easy to culture in future industrial applications.

### 2.3. Characterization and Identification of the Enzyme-Producing Strains

After isolation and screening, the DNA was extracted with a DNA kit (Omega Bio-tek, Inc., Ltd., Norcross, GA, USA) and then the 16S rRNA genes were amplified with polymerase chain reaction (PCR) using the pair of universal primers: 27F (5′-AGAGTTTGATCCTGGCTCAG-3′) and 1541R (5′-AAGGAGGTGATCCAGCCG CA-3′), according to previous research [33,34]. The PCR mixture (25 μL) consisted of 12.5 μL 2× Phanta Max Master Mix (Vazyme Biotech Co., Ltd., Nanjing, China), 9.5 μL ddH_2_O, 0.2 μm primers and 10 ng of template DNA. Thermocycling parameters included: initial denaturation at 98 °C for 1 min; 30 cycles (98 °C for 10 s, 50 °C for 30 s, 72 °C for 60 s); a final elongation at 72 °C for 5 min. The amplified products were detected using 1% agarose gel electrophoresis. The sequence determination was conducted by Sangon Biotechnology Company (Shanghai, China), and the strains were compared and detected using the tool’s (BLAST) website at the National Center for Biotechnology (NCBI) at https://novopro.cn/blast/blastn.html, accessed on July 1 2021. The nucleotide sequence data have been submitted to GenBank and the accession numbers are OK053813 to OK053818.

### 2.4. Insect Rearing

BSF eggs were maintained in a constant temperature (28 °C) incubator with a relative humidity of 60–70% for hatching [35]. Approximately 2 g of BSF eggs were laid on a sterile gauze grid on sterile boxes (120 × 90 × 80 mm) containing the relevant diet. After hatching, the BSF larvae fell into the sterile food waste and were dosed with six different bacterial culture solutions (A to G, wherein A is a control; Table 1). They were kept at 28 °C with a relative humidity of 60–70% and were fed with sterile food waste until 6 days old [35]. Seven diets were used in the current study: sterile food waste was used as a control; the other six groups were fed with sterile food waste dosed with *Lysinibacillus sphaericus*, *Proteus mirabilis*, *Citrobacter freundii*, *Pseudocitrobacter faecalis*, *Pseudocitrobacter anthropic*, and *Enterococcus faecalis* (labeled A to G, respectively (Table 1)). These strains were isolated from the previous steps. Each experimental group with the bacterial strains was inoculated with 20 mL of distilled water containing 10^8^ CFU/mL of bacterial cells, while 20 mL of distilled water without bacterial cells was added into the control group. The diets and bacteria were homogenized and the BSF larvae fell freely into them through the gaps in the gauze on each box for the 12 days spent in each container.

### 2.5. Sample Collection and Experimental Work

At least 500 samples were used per treatment group. Samples “A” could be collected for 7 days as a control (A1, A2, A3, A4, A5, A6, and A7), three larvae at each time (a total of 21 larvae were obtained), while the others were also taken over 7 consecutive days. A1 represents the gut microbiome of 6-day-old larvae fed with sterile FW. Meanwhile, B1, C1, D1, E1, and F1 samples were collected from 6-day-old larvae intestinal bacteria fed with sterile FW that was inoculated with *Lysinibacillus sphaericus*, *Proteus mirabilis*, *Citrobacter freundii*, *Pseudocitrobacter faecalis*, and *Pseudocitrobacter anthropi*, respectively. Samples “G” were fed with sterile food waste dosed with *Enterococcus faecalis*. After 6 days, the BSF larvae were too small to operate. Therefore, samples were taken from G3 (8-day-old larvae) and a total of 15 larvae were obtained from three replicates. Guts from three larvae per sample were pooled for DNA extraction, resulting in a total of three biological replicates per time point for each treatment. A total of 47 samples were subsequently submitted for sequencing.

### 2.6. DNA Extraction and 16S rRNA Sequencing

The DNA of all samples was extracted using E.Z.N.A.^®^ Bacterial DNA Kit (Omega Bio-tek, Inc., Ltd., Norcross, GA, USA), following the manufacturer’s protocol and normalized to equal concentrations before downstream processing. Before DNA extraction, the collected larvae were starved for 24 h to empty their gut to reduce the feed and excrement with non-symbiotic bacteria. For each batch, a total of 141 larvae from 47 sampling points were rinsed with sterile water after alcohol cleaning, and anatomized to extract the total DNA from the gut. The DNA products from three independent batches were mixed in equal concentrations and aliquoted for analysis.

The DNA samples were studied by sequencing the V3–V4 regions of bacterial 16S rRNA (Sangon Biotech, Shanghai, China) for the community profiling of the intestinal microbiotas using the 341 F (5′-CCTACGGGNGGCWGCAG-3′) and 805 R (5′-GACTACHVGGGTATCTAATCC-3′) primer set [36,37]. The PCRs were conducted using the following program: 3 min of denaturation at 94 °C, 25 cycles of 20 s at 94 °C, 20 s for annealing at 55 °C, followed by 30 s for elongation at 72 °C, and a final extension at 72 °C for 5 min. The PCR mixture (30 μL) consisted of 15 μL 2× Hieff^®^ Robust PCR Master Mix, sterilized ddH_2_O, 0.2 μm primers, and 10 ng of template DNA. The resulting PCR products were extracted from a 2% agarose gel and further purified using the Hieff NGS™ DNA Selection Beads (Yeasen Biotechnology Co., Ltd., Shanghai, China) and quantified using a Qubit3.0 Fluorometer (Invitrogen, Carlsbad, CA, USA), according to the manufacturer’s protocol. According to the standard protocols provided by Sangon Biotech (Shanghai, China) Co., Ltd., purified amplicons were pooled in equimolar concentrations and sequenced on an Illumina MiSeq platform (Illumina, San Diego, CA, USA) in PE300 mode.

### 2.7. Bioinformatics and Statistical Analyses

The bacterial 16S rRNA gene amplicon libraries were prepared using Illumina Miseq™ and were converted into sequenced reads using base-calling analysis. PRINSEQ (Argonne National Laboratory, Argonne, IL, USA) was employed to filter, reformat, and trim the genomic sequence data [38]. Low-quality sequences (*Q* < 20) were discarded. Quality trimming, chimera checking, singleton removal, and assignment of the obtained sequences to operational taxonomic units (OTUs) at 97% similarity level were conducted using Usearch v.11.0.667 [39]. The taxonomic affiliation of the resulting OTUs was identified by using RDP classifier [40]. A representative sequence for each OTU was filtered for deeper annotation. The microbial diversity, the standard diversity, and the richness indices (Ace, Chao1, and Shannon indices, respectively) were investigated using the Mothur software [41]. Intergroup comparisons were performed using a one-way analysis of variance (ANOVA), followed by Scheffe’s post hoc test (*p* < 0.05). STAMP software 6.0 [42] and R software 5.5 were adopted to analyze the bioinformatic sequence data sets. Downstream analysis was performed with Mothur [41] and the R package Agricolae (v.3.6.0). OTUs that were differentially abundant among the seven groups were assessed using the LEfSe approach [43].

## 3. Results

### 3.1. Isolation of Organic Compounds Degrading Bacteria

Cellulase-, protease-, and lipase-secreting bacteria were selected and isolated from the BSF larvae using selection media containing each organic compound. Six strains were identified by using 16S rRNA sequence alignment as follows: the strain for lipase and cellulase activities was identified as *Lysinibacillus sphaericus* with 100% identity; the three strains for protease activities were identified as *Enterococcus faecalis* with 99.79% identity, *Proteus mirabilis* with 99.79% identity, and *Citrobacter freundii* with 99.72% identity; two strains for cellulase activities were identified as *Pseudocitrobacter faecalis* with 99.58% identity and *Pseudocitrobacter anthropi* with 99.93% identity (Table 1).

To obtain the optimal pH for bacteria to grow, the strains were grown in LB medium with a pH between 2.0 and 8.0. There were three replicates for each strain and each pH value. Alternatively, the pH for *Lysinibacillus sphaericus* growth was 2.0 to 8.0. As shown in Figure 1, *Lysinibacillus sphaericus* grew better under alkaline conditions. The strong acid environment was more suitable for *Pseudocitrobacter faecalis* growth, while the other strains were more adaptable to a neutral environment (Figure 1).

### 3.2. Taxonomic Composition of Bacterial Communities

The resulting libraries contained an average of 86,139 16S rDNA sequences per sample, with an average length of 428 bases. Out of the bacterial phyla that were identified (Figure 2A), Proteobacteria, Firmicute, and Bacteroidetes were the most dominant phyla that were associated with BSF of all samples. The phyla of the bacterial communities in Group A consisted of Proteobacteria (83.80%), Firmicutes (14.00%), and Bacteroidetes (0.66%). The phyla that were found in Group B were Proteobacteria (82.66%), Firmicutes (16.75%), and Bacteroidetes (0.18%). The microorganism compositions of the other groups with other strains are listed in Appendix A.

Out of the bacterial genera that were identified (Figure 2B), 19 bacterial genera that shared the seven larval groups were identified. *Ignatzschineria*, *Providencia*, *Proteus*, *Klebsiella*, and *Vagococcus* were the most dominant genera that were associated with the BSF larvae of all samples. In Group A, the proportions of these five bacteria represented 49.69, 14.93, 9.20, 5.02, and 3.08% of the total, respectively (Appendix A). The contents of *Proteus*, *Morganella*, *Bacillus*, *Paenalcaligenes*, and unclassified *Enterococcaceae* in Group B were 12.3, 5.01, 1.09, 1.63, and 1.61%, respectively, which were all higher than the blank control (Group A) and other groups. In Group C, the top five strains were *Ignatzschineria*, *Providencia*, *Proteus*, *Vagococcus*, and *Klebsiella*. *Ignatzschineria* and *Providencia* were the two most dominant strains in all of the experimental groups. In Group D, *Enterococcus* was the third most abundant bacteria, rather than *Proteus* (as was the case in the other groups). In addition, the number of *Ignatzschineria* and *Enterococcus* in Group D was the largest among all groups. Furthermore, the abundance of *Proteus* in Group F was the highest.

### 3.3. Differences in Intestinal Bacterial Communities over Time

At the genus level, *Ignatzschineria* was the least abundant among the seven groups on the first day of sampling. In the first two days of sampling, *Proteus* and *Klebsiella* were the main components of the intestinal bacteria (Figure 3). As feeding days changed, the number of *Ignatzschineria* in Group B and D showed a rising trend from day 1 to day 4, and then decreased to day 7. Alternatively, it demonstrated the same trend in the first five days in other groups as it did in the previous two groups, but there was a rapid increase on the sixth and seventh days, until it reached a maximum on the seventh day.

Using Welch’s *t*-test and ANOVA significant difference analysis, the six groups of colonizing strains were compared with the blank experimental Group A. The comparison found that the amount of the other five strains did not change significantly, with the exception of *Lysinibacillus sphaericus* in Group B—*Lysisnbacillus* in Group B was 0.66% and, in the other groups, it was below 0.07% (Appendix A). At the same time, the characteristics of the bacteria were found to be significantly different between the seven groups (Figure 4) in the linear discriminant analysis effect size (LEfSe) analysis. As there was no significant difference between Group D and G and the control group, it is not shown in Figure 4. Consistent with the results of Welch’s *t*-test significant difference analysis, *Lysinibacillus sphaericus* was found to be a particular component of Group B. Based on 16S rRNA sequencing and OTU contributions from the abundant phyla and others for facultative, anaerobic, and aerobic bacteria according to BugBase analysis, the microorganisms in the larvae gut were revealed (Figure 5). Most of the genes were obtained on aerobic and facultatively anaerobic bacteria. Generally, facultative anaerobes can grow under conditions with or without oxygen, but prefer aerobic conditions.

## 4. Discussion

In this study, six functional strains were isolated in total: *Lysinibacillus sphaericus*, *Proteus mirabilis*, *Citrobacter freundii*, *Pseudocitrobacter faecalis*, *Pseudocitrobacter anthropi*, and *Enterococcus faecalis*. The colonization experiments showed that *Lysinibacillus sphaericus* may be grown and accumulated effectively in the gut of BSF larvae (Figure 3). In other words, the colonization method we adopted was proven to be a potential method for certain microbes.

*Lysinibacillus sphaericus* (formerly known as *Bacillus sphaericus*) is a Gram-positive, spore-forming bacterium that has been used in the biological control of mosquitoes and bioremediation [44,45], and was initially isolated from an adult black fly in Nigeria [46]. *Lysinibacillus fusiformis* was previously isolated from the eggs of the BSF colony, and could dominate the larval microbiota and increase larval weight and survival [30]. *Lysinibacillus sphaericus* can be found in a variety of environments, such as soil, water, and animal intestines. Chantarasiri et al. [47] reported that the *Ligninolytic bacterium* JD1103 was isolated from soil samples that were collected from the wetland ecosystems in Rayong Province, Thailand and was identified as *Lysinibacillus sphaericus* JD1103 based on 16S rRNA sequence analysis. Similarly, it was reported that lignin-degrading bacteria are not well understood. In their research, an effective lignin-decomposing bacterial strain, BR2308, was isolated from the coastal wetland ecosystem and named *Lysinibacillus sphaericus* BR2308 [48]. Most reports showed that *Lysinibacillus sphaericus* can produce insecticidal proteins that have high activity against mosquito larvae [49,50]. However, it is one of the functional cellulase-producing strains that we screened in the BSF larvae gut using CMC-Na solid medium. Others have isolated cellulose-producing *Lysinibacillus sphaericus* MTCC No. 9468 from the gut of earthworms (*Eisenia foetida*) [51].

Several researchers have found that environmental and feeding sources have a significant influence on the overall composition of the insect microbial community [14], such as *Hermetia illucens* intestinal microbiota [28,52,53] and *Drosophila melanogaster* gut microbiota [54]. Thus, before sequencing data analysis, it was supposed that all of the group samples would have more target bacteria in them because the larvae were fed sterile food waste with different target bacteria. The inoculation of the substrate with substrate-associated microorganisms affected larval performance and caused major changes in larval and substrate microbiota, whereas egg-associated microorganisms did not influence performance [30]. De Smet et al. [55] found that BSF larvae can exhibit delayed performance on sterile substrates and some bacteria participate in intestinal digestion and nutrient utilization. However, the substrate has a significant effect on larval gut bacterial community composition.

To reduce the external influence, larvae that were fed with autoclaved food waste were used as the control in this study. Yang et al. [56] found that the BSF larval gut microbial community structure was significantly influenced by starving, even over a short time (e.g., 24 h). To reduce the influence of food residues in the intestine on intestinal microorganisms, according to a recent study [57], the collected larvae were starved for 24 h to allow for egestion of their ingested contents before DNA extraction.

Interestingly, after conducting colonization experiments of six functional strains, only *Lysinibacillus* in Group B showed a significant increase in bacterial communities. The others were similar to the blank control group (Group A), and the gut microbiome remained more stable. In particular, the content of *Lysinibacillus sphaericus* in Group B was found to account for 0.66% of the total in all sampling sites, which was significantly higher than the other groups (*p* < 0.05). Bacteria serve directly as food for fly larvae and help decompose macronutrients [10]. Such decomposition of microbes via gut-based mechanisms (including pH, enzymes, and antimicrobial proteins) can explain the selective inactivation of microbes, as reported for fly larvae [10]. *Escherichia coli* and *Bacillus subtilis* through the gut passage were identified as having been completely inactivated by using the fluorescent bacteria, and *Enterococcus faecalis* for BSF larvae caused low reductions in both the larvae and residue [10]. Microbe inactivation by fly larvae depends on the specific microbe and strain. Microbes that survive the gut passage are candidates for application. Others reported that microbial colonization depends on different physico-chemical conditions in the lumen of different gut compartments, which shows extreme changes in pH and oxygen availability [14]. The optimum pH for the growth of *Lysinibacillus sphaericus*, *Proteus mirabilis*, *Citrobacter freundii*, *Pseudocitrobacter faecalis*, *Pseudocitrobacter anthriopi*, and *Enterococus faecalis* was approximately 8.0, 6.0, 7.0, 4.0, 7.0, and 6.0, respectively. It can be seen from the above data that *Lysinibacillus sphaericus* is alkali tolerant. In addition, after the experiment, it was found that *Lysinibacillus sphaericus* is aerobic and the others are facultative anaerobes. The results of 16S rRNA sequencing predicted that the aerobic bacteria in the gut of BSF larvae account for more than 50% (Figure 5). According to the previous research, the anterior region of the mid-gut of BSF larvae has an acid luminal content, the mid-region presents a strongly acidic pH, and the posterior region has an alkaline luminal content [28]. The environment of the posterior region allows *Lysinibacillus sphaericus* to grow well. In addition, although many insects will shed the exoskeletal lining of the foregut and hindgut during their molts, most of the changes do not cross into the space that is adjacent to midgut epithelial cells [14]. Moreover, many insect guts display specialized crypts or paunches that promote microbial persistence [14]. Taking the evidence into account, it was inferred that that this may be the reason that *Lysinibacillus sphaericus* could colonize in the gut of the larvae, while the other strains cannot.

Proteobacteria, Firmicutes, and Bacteroidetes were the dominant phyla (Figure 2) in the larval gut, which is consistent with the main taxa of the intestinal microbes of BSF that have been found elsewhere. Others have assessed the effects of different diets and their microbial community on the mid-gut microbiota of BSF larvae and found that, at the phylum level, Proteobacteria, Firmicutes, Bacteroidetes, and Actinobacteria were the most important microbes in the gut [28]. In previous studies, Actinobacteria, Firmicutes, Bacteroidetes, and Proteobacteria were also found to represent the most abundant phylum-level compositions in the gut of larvae [10,53]. The dominance of bacterial communities among the groups of BSF larvae that were fed different diets showed that the type of food and the feeding time could slightly influence bacterial diversity (Figure 2). Previous studies show that the dominating genera (mainly *Bacterioides*, *Dysgonomonas*, *Morganella*, *Enterococcus*, *Providencia, Klebsiella*, and *Bacillus*) have a large variability between BSF larvae and studies, which is likely due to the variability in diet [14,53,58,59]. In the present work, similar results were found in that *Ignatzschineria*, *Providencia*, *Proteus*, *Klebsiella*, and *Vagococcus* were in the top five at the genus level. Compared to the intestinal microorganisms of other insects, these bacterial communities are unique [60]. Chaobing Luo et al. [61] discovered that *Lactococcus*, *Enterococcus*, *Bacillus*, *Citrobacter*, *Vagococcus*, and *Serratia* are the main components of the bamboo snout beetle *Cyrtotrachelus buqueti* intestinal bacteria. One et al. [62] found that, in mealworms that were fed with wheat bran, the majority population in the mealworm gut included *Enterococus*, *Clostridium*, *Erwinia*, and *Lactococcus*. In addition, in the mid-gut of *Aedes aegypti*, *Pseudomonas*, *Pseudoalteromonas*, *Luteibacter*, and *Rhodanobacter* were the most abundant and dominant populations [63]. The composition and abundance of bacterial communities in the gut of insects are not only affected by diet, but also by the species of insect. Therefore, it is important for us to study the composition and function of the intestinal microorganisms of BSF larvae. In this study, the structure of the intestinal microorganisms of BSF larvae was scarcely affected by the colonization of functional strains.

Others have reported that exogenous bacterial inoculation oxidized fiber and assisted the joint action of BSF larvae and the gut microbiome, thereby increasing the rate of bioconversion [64]. Others have conducted similar experiments with endogenous bacteria to assess the co-conversion efficacy in BSF larvae after the development of soybean curd residues with *Lactobacillus buchneri* [65]. They inoculated *Lactobacillus buchneri* into the feed for fermentation and then used it to feed the BSF larvae to study the changing parameters, such as the reduction rate of dry matter and the biotransformation rate after feeding. On the contrary, in order to better digest, decompose, and utilize various organic wastes by BSF larvae, a co-conversion technique was established. The result provides a practical and promising method for converting organic wastes into BSF larval biomass.

## 5. Conclusions

The functional bacterial strains that were derived from the intestinal tract of the insects were isolated and screened. We then enriched and cultivated the specimens in vitro, and then colonized the intestines of BSF larvae. The colonization effect was evaluated based on changes in the abundance of intestinal microorganisms after sequencing. Endogenous bacteria were used to construct a synergistic transformation system of ‘BSF larvae-functional microorganisms’ through the ‘directed transformation’ of the intestinal microorganisms of BSF larvae. We inoculated bacteria and fed the BSF larvae at the same time to determine the effects of endogenous bacterial agents on the intestinal microorganisms of BSF larvae. Finally, *Lysinibacillus sphaericus* may be allowed to colonize the gut of BSF larvae. The results indicate that the inoculation method may be a potential pathway to transform the intestinal microorganisms of the BSF directionally. Further research (e.g., of the gastrointestinal tract) is needed to investigate the cause of the partial variability of intestinal bacteria and the colonization of some bacteria.

## Figures and Tables

**Figure 1 animals-11-03475-f001:**
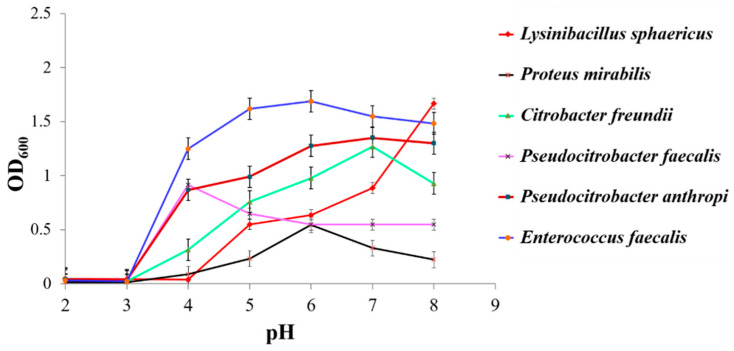
The influences of pH on the growth of different strains. The *X*-axis indicates different pH values and the *Y*-axis denotes the absorbance of visible light at 600 nm. The greater the absorbance, the better the growth of the strain. The error bars were derived from the standard deviation between replicates (*n* = 3).

**Figure 2 animals-11-03475-f002:**
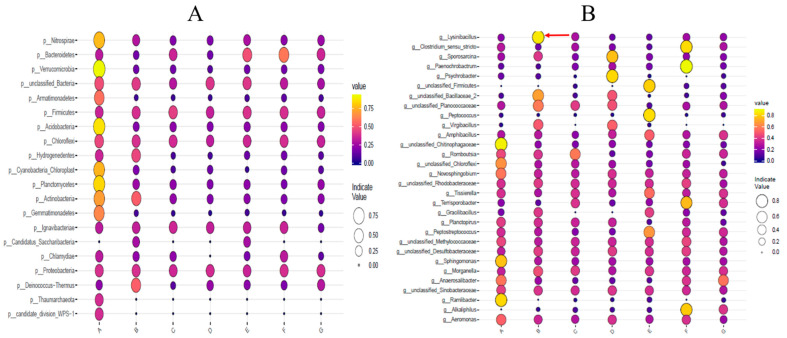
Relative abundance of taxa at phylum level (**A**) and genus level (**B**). The *X*-axis represents the experimental group, and the *Y*-axis denotes the taxonomic classification information. In the value label, the yellower and larger the bubble, the higher the species abundance. The red arrow in (**B**) indicates the target strain *Lysinibacillus* added to Group B.

**Figure 3 animals-11-03475-f003:**
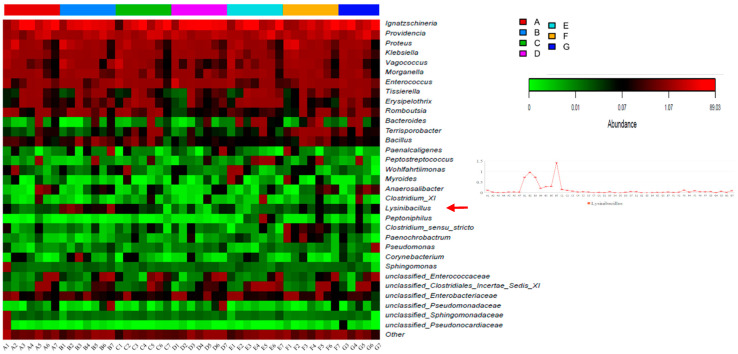
Heatmap based on microbiota abundance at genus level. Each row represents bacteria at genus level, and each column represents a sample. The color of each cell denotes the relative abundance of the species in the sample. The darker the color (red), the higher the abundance of the species. The red arrow and line chart indicate the target strain *Lysinibacillus* added to group B.

**Figure 4 animals-11-03475-f004:**
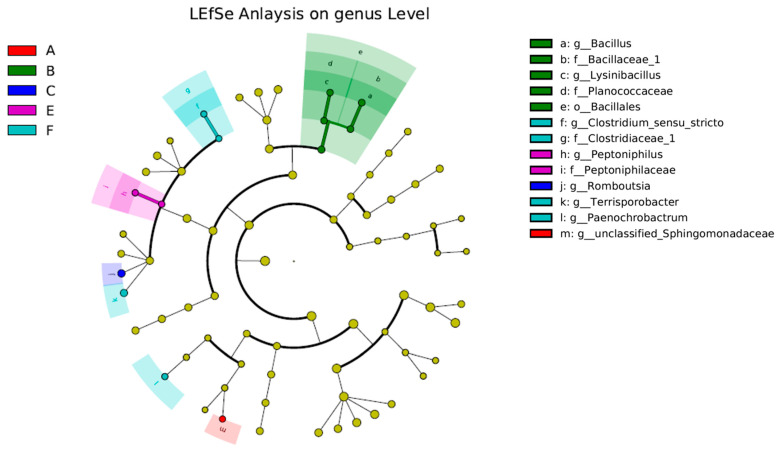
Differential abundance microbial cladogram obtained using LEfSe. Different colors indicate different groups. The nodes of different colors in the branches represent the groups of microorganisms that play an important role in the corresponding group of the color, and the yellow nodes indicate the groups of microorganisms that have not played an important role. The species names in Roman alphabet are shown in the legend on the right.

**Figure 5 animals-11-03475-f005:**
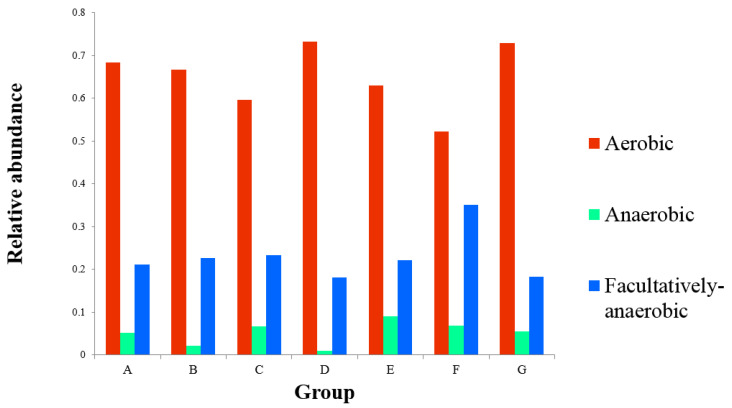
Bar graph of oxygen demand in each group. The *X*-axis refers to the different groups. The *Y*-axis indicates the relative abundance of different strains in each group.

**Table 1 animals-11-03475-t001:** Description of experimental groups and GenBank accession numbers.

Group	Base Pairs	Inoculum (Closest GenBank Relative)	Closest Relative Accession No.	Identity %	Phyla	Genus	Genus Submission
A	/	None	/	/	/	/	/
B	1268	*Lysinibacillus sphaericus*	MT279445.1	100	Firmicutes	*Lysinibacillus*	OK053815
C	1439	*Proteus mirabilis*	KC456557.1	99.79	Proteobacteria	*Proteus*	OK053814
D	1442	*Citrobacter freundii*	MH973163.1	99.72	Proteobacteria	*Citrobacter*	OK053817
E	1446	*Pseudocitrobacter faecalis*	KF057942.1	99.58	Proteobacteria	*Pseudocitrobacter*	OK053813
F	1447	*Pseudocitrobacter anthropic*	NR_125691.1	99.93	Proteobacteria	*Pseudocitrobacter*	OK053816
G	1462	*Enterococcus faecalis*	MT544896.1	99.79	Firmicutes	*Enterococcus*	OK053818

## Data Availability

All data are provided in full in Section 3 of this paper, apart from the DNA sequence encompassing the larvae intestinal bacteria gene cluster, which is available at www.ncbi.nlm.nih.gov/genbank/ under GenBank Submission OK053813 to OK053818. The raw data from the next-generation sequencing platform were submitted to the Sequence Read Archive (SRA) at https://submit.ncbi.nlm.nih.gov/subs/sra/ under BioProject ID PRJNA761527.

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
