# Peer review of "Directional Changes in the Intestinal Bacterial Community in Black Soldier Fly (Hermetia illucens) Larvae"

_animals, 2021, doi:10.3390/ani11123475_

Round 1
Reviewer 1 Report
General comments
The manuscript entitled “Directional changes in the intestinal bacterial community in black soldier fly (Hermetia illucens) larvae” describes an experiment aiming to assess changes in bacterial communities of the gut of black soldier flies under different diets with different bacterial culture solutions. The results on the taxonomic composition of the bacterial communities of this important fly species, that is used for bioconversion of organic waste, are relevant. However, it is not clear if there were replicates of the experimental group and how many. The only reference to these were in line 135 (“The DNA products from three replicate samples...”). If these were indeed replicates of the different diets, it should be explicitly stated in the methods and the results for each replicate should be provided. Also, another concern is the results on abundance of bacterial taxa. It is not explained how these were obtained from amplicons, if even possible. These two main aspects need to be fully explained in order to understand if the conclusions are supported.
Detailed comments:
Line 12 – The sentence “That fly larvae cohabit in some form of symbiosis with microbes.” needs to be reformulted. To which larvae does it refer to ?
Line 15 - “the residue” refers to what? Organic waste?
Line 18 to 20 – the sentence is not clear. Do you mean that the communities could be affected by addition of certain bacteria during rearing?
Line 52 to 55 – citations are missing about the mentioned works on the mammalian gut bacteria and screening of high-efficiency enzyme producing strains
Lines 66 to 70 – If these sentences refer to the objectives and main results of this study, they should be reformulated to clearly state that. For example: “In this study we targeted… “, “we inferred...”. A short sentence about the procedures used in this study to achieve the results is also missing, in order to understand the description of the methods that follow.
Line 68 - What does it mean “...degrading effect on various wastes will be discarded later?”. Also, the last sentence “Changes in the conversion rate of biodegradation provide a theoretical basis for this research” is not clear and does not fit in the context. If this is relevant, it should be explained.
Methods
- The number of samples used in the experiments has to be clearly stated. How many larvae were used for the isolation and screening of enzyme-producing strains? Was this done in pools of larvae?
Line 97 – If these primers are already published, a citation is missing. If they were designed for this study, explain how and which software was used. What were the PCR conditions and reagent concentrations used?
Line 101 – by using BLAST (NCBI)? A citation is missing.
Line 103 – is there a previous work explaining the rearing conditions of the eggs? If there is, it should be cited.If not, further details should be given.
Line 104 – “Some 2 g” is “approximately 2 g”? Can you state the approximate number of eggs that corresponds to?
Line 112 - “were homogenised”?
Table 1 – The Diet column can be removed. It would be informative to include the Phyla or Family to which each of the bacteria species belong to. Explain the Identity column in the legend of the table
Line 116 – indicate how many larvae were collected at each day of sampling
Line 118 – How many replicates of each diet?
Line 124 – explain why was the G3 sample treated this way
Line 135 – are these three replicate samples within each of the seven replicates of each of the seven experimental groups? Please detail.
Line 139 – If these primers are already published, a citation is missing. If they were designed for this study, explain how and which software was used.
Lines 142-143 – The final concentration of each reagent should be given instead of the volumes
Lines 151 to 155 – too long sentence. Please split it and detail how the structure analyses of large-scale phyla and community complexity analyses were done. All the filtering and data analysis can go to the next section that can be named “Bioinformatics and statistical analyses”
Line 165 – how were the abundances estimated?
Lines 177 – refer to Table 1
Line 178 – how many replicates were used for each species and each pH?
Line 181 – Citrobacter freundii does not seem to do particularly well in acidic conditions (green line in figure 1), but Pseudocitrobacter faecalis does.
Fgure 1 – what represent the bars? Standard deviations? It is better to indicated confidence intervals. How many replicates were analysed?
Figure 2 – line 207 “experimental group” instead of “sample”; line 208 “taxonomic classificatio” instead of “species classification”; the colour is not needed in this figure, as the size of the bubbles show the same information
Line 215 – the trend is not clear from the heatmap in figure 3. Another figure could be more informative with a line plot of abundance versus time for each of the species showing a significant temporal trend.
Figure 3 – no red arrow is visible.
Line225 – The results of the tests should be given as supplementary material. Figure 4 is not necessary because it is very redundant (same values of proportion repeated several times for each group). In the text you could state the proportion of Lysisnbacillus in group B (0.7 %) and in the other groups (below 0.1?)
Line 271 – it is not clear what the 73.02 % refers to.
Line 275 – 276 – these values do not match with figure 1.
Figure 6 should be included in the results.
References need to be corrected. They have no journal names. Authors lists need to be checked and corrected.
Author Response
Replies to comments on animals-1438858
General comments: The manuscript entitled “Directional changes in the intestinal bacterial community in black soldier fly (Hermetia illucens) larvae” describes an experiment aiming to assess changes in bacterial communities of the gut of black soldier flies under different diets with different bacterial culture solutions. The results on the taxonomic composition of the bacterial communities of this important fly species, that is used for bioconversion of organic waste, are relevant. However, it is not clear if there were replicates of the experimental group and how many. The only reference to these were in line 135 (“The DNA products from three replicate samples...”). If these were indeed replicates of the different diets, it should be explicitly stated in the methods and the results for each replicate should be provided. Also, another concern is the results on abundance of bacterial taxa. It is not explained how these were obtained from amplicons, if even possible. These two main aspects need to be fully explained in order to understand if the conclusions are supported.
Response:Thank you for your comments. We carefully checked the full text as you suggested. Based on reading and summarizing literatures massively, we screened and replaced some inaccurate literature, the purpose to make a better connection with this article. The details are as follows.
Detailed comments:
- Line 12 (in the original manuscript) -The sentence “That fly larvae cohabit in some form of symbiosis with microbes.” needs to be reformulted. To which larvae does it refer to ?
Reply: Thanks for your kind reminder. We have replaced “That” by “Insect”. (See page 1, line 12)
- Line 15 (in the original manuscript) - “the residue” refers to what? Organic waste?
Reply: Thanks very much for your professional suggestion. Yes, “the residue” refers to the “organic waste – food waste”. In the revision, we have replaced “the residue” by “organic waste” thoughout the text in the revision. (See page 1, line 15)
- Line 18 to 20 (in the original manuscript) - the sentence is not clear. Do you mean that the communities could be affected by addition of certain bacteria during rearing?
Reply: Thank you for your comments. Yes, the communities could be affected by addition of certain bacteria during rearing. De Smet, et al. [1]found that feed-associated bacteria can affect growth and development of H. illucens larvae either positively or negatively. H. illucens larvae do not develop rapidly on sterile substrates, suggesting that some bacteria are beneficial and even essential for nutrient utilization[2]. Microbes that survive the gut passage are candidates for microbes that contribute more to larval development, besides just being a nutrient source. Bacteria present in feed substrates could also compete with H. illucens larvae for nutrients[3]. (See page1, line 18-20)
- De Smet, J.; Wynants, E.; Cos, P.; Van Campenhout, L. J. A.; Microbiology, E., Microbial community dynamics during rearing of black soldier fly larvae (Hermetia illucens) and impact on exploitation potential. Appl Environ Microb. 2018, 84, (9), e02722-17. APPL ENVIRON MICROB
- Cifuentes, Y.; Glaeser, S. P.; Mvie, J.; Bartz, J.-O.; Müller, A.; Gutzeit, H. O.; Vilcinskas, A.; Kämpfer, P. J. A. V. L., The gut and feed residue microbiota changing during the rearing of Hermetia illucens larvae. Antonie van Leeuwenhoek. 2020, 113, (9), 1323-1344.
- Gold, M.; Tomberlin, J. K.; Diener, S.; Zurbrügg, C.; Mathys, A. J. W. M., Decomposition of biowaste macronutrients, microbes, and chemicals in black soldier fly larval treatment: A review. Waste Manage. 2018, 82, 302-318.
- Line 52 to 55 (in the original manuscript) - citations are missing about the mentioned works on the mammalian gut bacteria and screening of high-efficiency enzyme producing strains.
Reply: Thanks for your kind reminder. We have added the related references. (See page 2, line 58, line 62)
- Lines 66 to 70 (in the original manuscript) -If these sentences refer to the objectives and main results of this study, they should be reformulated to clearly state that. For example: “In this study we targeted… “, “we inferred...”. A short sentence about the procedures used in this study to achieve the results is also missing, in order to understand the description of the methods that follow.
Reply: Thanks very much for the valuable suggestion, in the revision we have rephrase the sentence as you suggested. (See page 2, line 82-88)
- Line 68 (in the original manuscript) - What does it mean “...degrading effect on various wastes will be discarded later?”. Also, the last sentence “Changes in the conversion rate of biodegradation provide a theoretical basis for this research” is not clear and does not fit in the context. If this is relevant, it should be explained.
Reply: Thanks very much for the valuable suggestion, in the revision we have rephrase the sentence as you suggested. (See page 2, line 82-88)
Methods
7. Line 76 (in the original manuscript) - The number of samples used in the experiments has to be clearly stated. How many larvae were used for the isolation and screening of enzyme-producing strains? Was this done in pools of larvae?
Reply: Thanks very much for the valuable suggestion, in the revision we have rephrase the sentence as you suggested. The number of samples clearly stated, and the ten intestinal samples were mixed and homogenized. (See page 3, line 96-100)
- Line 97 (in the original manuscript)- If these primers are already published, a citation is missing. If they were designed for this study, explain how and which software was used. What were the PCR conditions and reagent concentrations used?
Reply: Thanks very much for your comment. The references of the universal primers has been cited. (See page 3, line 121) The PCR conditions and reagent concentrations has been added. (See page 3, line 121-124)
- Line 101(in the original manuscript)- by using BLAST (NCBI)? A citation is missing.
Reply: Thanks very much for your comment. We have added the citation in the revision. (See page 3, line 127-130)
- Line 103 (in the original manuscript)- is there a previous work explaining the rearing conditions of the eggs? If there is, it should be cited.If not, further details should be given.
Reply: Thanks very much for your comment. In the revision, we have added the references citation. (See page 3, line132-133)
- Line 104 (in the original manuscript) - “Some 2 g” is “approximately 2 g”? Can you state the approximate number of eggs that corresponds to?
Reply: Thanks very much for your kind suggestion, we have replaced “some” by “approximately”. (See page 3, line 133)
- Line 112 (in the original manuscript) - “were homogenised”?
Reply: Thanks very much for your kind reminder, we have revised it in the revision. (See page 3, line 144)
- Table 1 - The Diet column can be removed. It would be informative to include the Phyla or Family to which each of the bacteria species belong to. Explain the Identity column in the legend of the table
Reply: Thanks very much for your kind reminder. We have rephrased the table 1 as your valuable suggestion. Table 1. Description of experimental groups and Genus Submission numbers. (See page 4, line 147 and Table 1)
- Line 116 (in the original manuscript) - indicate how many larvae were collected at each day of sampling.
Reply: Thanks very much for your kind suggestion, in the revision we have added the collected sampling numbers according to the suggestion. (See page 4, line150-152).
- Line 118 (in the original manuscript) - How many replicates of each diet?
Reply: Thanks very much for your kind suggestion, in the revision we have added the collected sampling numbers according to the suggestion. (See page 4, line59-161).
- Line 124 (in the original manuscript) - explain why was the G3 sample treated this way.
Reply: Thanks very much for your kind reminder. In the revision, we have added the explaintion according to the suggestion. (See page 4, line 157-159).
- Line 135 (in the original manuscript) - are these three replicate samples within each of the seven replicates of each of the seven experimental groups? Please detail.
Reply: Thanks very much for your kind reminder. In the revision, we have added the explaintion according to the suggestion. (See page 4, line 169-170).
- Line 139 (in the original manuscript) - If these primers are already published, a citation is missing. If they were designed for this study, explain how and which software was used.
Reply: Thanks very much for your comment. The references of the universal primers has been cited. (See page 4, line 174)
- Lines 142-143 (in the original manuscript) - The final concentration of each reagent should be given instead of the volumes.
Reply: Thanks very much for the reminder. In the revision, we have revised it in the manuscript. (See page4, line177-178)
- Lines 151 to 155 (in the original manuscript) - too long sentence. Please split it and detail how the structure analyses of large-scale phyla and community complexity analyses were done. All the filtering and data analysis can go to the next section that can be named “Bioinformatics and statistical analyses”
Reply: Thanks very much for your kind reminder. We have rephrased the paragraph as your valuable suggestion. (See page 5, line 195-203)
- Line 165 (in the original manuscript) - how were the abundances estimated?
Reply: Thanks very much for your comment. The microbial diversity, the standard diversity and richness indices (Ace index, Chao1 index and Shannon index, respectively) was investigated by using Mothur software. (See page 5, line 195-203)
- Lines 177 (in the original manuscript) - refer to Table 1
Reply: Thank you for your kind reminder, we have added the “Table 1” in the revision. (See page 5, line 213)
- Line 178 (in the original manuscript) - how many replicates were used for each species and each pH?
Reply: Thank you for your kind reminder, we have added the sentence “There were three replicates for each strain and each pH value” in the revision. (See page 5, line 215)
- Line 181 (in the original manuscript) - Citrobacter freundii does not seem to do particularly well in acidic conditions (green line in figure 1), but Pseudocitrobacter faecalis does.
Reply: Thank you for your kind reminder, we have added the sentence “And the strong acid environment was more suitable for Pseudocitrobacter faecalis growth.” in the revision. (See page 5, line 218-219)
- Fgure 1 - what represent the bars? Standard deviations? It is better to indicated confidence intervals. How many replicates were analysed?
Reply: Thank you for your kind reminder, we have added the sentence “The error bars derived from the standard deviation between replicates (n=3)” in the revision. (See page 6, line 223-224)
- Figure 2 - line 207 “experimental group” instead of “sample”; line 208 “taxonomic classificatio” instead of “species classification”; the colour is not needed in this figure, as the size of the bubbles show the same information.
Reply: Thanks very much for your kind reminder. We have rephrased the paragraph as your valuable suggestion. (See page 7, line 247-249)
- Line 215 (in the original manuscript) - the trend is not clear from the heatmap in figure 3. Another figure could be more informative with a line plot of abundance versus time for each of the species showing a significant temporal trend. Figure 3- no red arrow is visible.
Reply: Thanks very much for the suggestion. Because of the color difference, we have to look carefully, and the red arrow and a line chart has been added. (See page 7, figure 3)
- Line 225 (in the original manuscript) - The results of the tests should be given as supplementary material.
Reply: Thanks very much for the suggestion. Based on the accurate eta-squared values from multifactor analysis of variance (ANOVA) designs, as shown in the next table, the other five strains did not change significantly in top 17, so the different in mean proportions wan not shown. (See page 8, line 266)
No |
Genes |
Eta-Squared |
p-value |
q-value |
1 |
g__Lysinibacillus |
0.66 |
4.49E-08 |
1.20E-05 |
2 |
g__Sporosarcina |
0.61 |
4.04E-07 |
5.38E-05 |
3 |
g__Clostridium_sensu_stricto |
0.47 |
0.000134543 |
0.011929462 |
4 |
g__Terrisporobacter |
0.44 |
0.000349207 |
0.023222293 |
5 |
g__unclassified_Bacillaceae_2 |
0.38 |
0.002372294 |
0.12620603 |
6 |
g__Alkaliphilus |
0.36 |
0.004483993 |
0.198790342 |
7 |
g__Amphibacillus |
0.35 |
0.005020602 |
0.190782871 |
8 |
g__Namhaeicola |
0.34 |
0.007663849 |
0.254822963 |
9 |
g__Virgibacillus |
0.32 |
0.011339611 |
0.335148516 |
10 |
g__unclassified_Chloroflexi |
0.31 |
0.013953056 |
0.371151284 |
11 |
g__Rhodoligotrophos |
0.29 |
0.021031564 |
0.50858145 |
12 |
g__Clostridium_XlVa |
0.29 |
0.02262241 |
0.501463431 |
13 |
g__Gp17 |
0.29 |
0.024918095 |
0.509862568 |
14 |
g__unclassified_Desulfuromonadaceae |
0.29 |
0.025299825 |
0.480696669 |
15 |
g__Paenochrobactrum |
0.28 |
0.027924242 |
0.495189886 |
16 |
g__unclassified_Clostridiales_Incertae_Sedis_XI |
0.28 |
0.032106942 |
0.533777906 |
17 |
g__unclassified_Firmicutes |
0.26 |
0.049022518 |
0.767058219 |
- Figure 4 is not necessary because it is very redundant (same values of proportion repeated several times for each group). In the text you could state the proportion of Lysisnbacillus in group B (0.7 %) and in the other groups (below 0.1?)
Reply: Thanks very much for the valuable suggestion, in the revision we have changed the figure4 as supplementary Figure S1, and rephrase the sentence as you suggested: Lysisnbacillus in group B was 0.66 % and in the other groups was below 0.1%. (See page 8, line 267-268)
- Line 271 (in the original manuscript) - it is not clear what the 73.02 % refers to.
Reply: Thanks for your kind reminder. The original expression was misleading; we have replaced the references by “the content of Lysinibacillus sphaericus in Group B was found to account for 0.66% of the total in all sampling sites”. (See page 9, line 329-330)
- Line 275 - 276 (in the original manuscript) - these values do not match with figure 1.
Reply: Thanks for your kind reminder. The original expression was misleading; we have replaced the references by “was about 8.0, 6.0, 7.0, 4.0, 7.0, and 6.0 respectively”. (See page 9, line 342)
- Figure 6 should be included in the results.
Reply: Thanks for your kind reminder. In the revision, we have added the sentence according to the suggestion. (See page 8, line 273-278)
- References need to be corrected. They have no journal names. Authors lists need to be checked and corrected.
Reply: Thanks for your kind reminder. We have checked the entire references and corrected in the revision. (See page 12-15, line 423-590)

Reviewer 2 Report
In this study, the authors isolated six different strains from guts of black soldier fly larvae (BSFL) and used these strains to inoculate sterilized food wastes. Their aim was to see whether the BSFL gut microbiota can be enriched and thus shaped by adding certain functional bacterial isolates. In addition, they wanted to determine which isolated strains can establish themselves within the larval gut. The idea of improving larval digestion by adding microbes with certain beneficial features that could help to degrade hardy polysaccharides like lignin or cellulose is very interesting. However, as the authors already state in their summary, they are only presenting the results from a preliminary experiment. Thus, the spectrum of applied methods is rather narrow and does not allow for very detailed conclusions as it was attempted by the authors. Using 16S amplicon sequencing without additional quantification or detection methods is not enough to sufficiently verify the enrichment of certain species, let alone strains. With this in mind, I suggest that the conclusions should be kept more moderate and descriptive. The discussion could also be improved by referring to more fitting and topical literature on the BSF gut microbiota.
As the data comes from a preliminary study, another option would be to convert the submission into a short communication.
Major comments
- Language needs to be improved throughout the manuscript. Some parts of the manuscript are not easy to understand
- Line 72: Please add additional information on the rearing conditions of the BSF such as temperature, relative humidity, substrate that was used for rearing, container sizes, larval densities, etc.
- Material and methods should be restructured. Currently, related information is distributed across different sections
- Accession number (PRJNA761527) for the publicly deposited sequencing data does not work. Please make the data publicly available by re-uploading or changing the privacy settings of your project
- You did not include a “positive control” with food waste that was not sterilized. How did the sterilization of food waste affect larval development? From experience and other published studies (e.g. https://doi.org/10.1093/femsec/fiab054) we know that especially neonate larvae rely on the substrate microbiota and their development is severely inhibited if the substrates are sterilized. The missing “positive control” and the relevance of the substrate microbiota should be included in the discussion.
- Lines 336-338: Relying on 16S amplicon data alone to make such an assumption is rather ambitious. I’m missing additional (quantitative) methods to verify the gut colonization with the proposed isolates or more specifically with Lysinibacillus sphaericus.
- How do you explain that Lysinibacillus in Group B was found via 16S amplicon sequencing while the other supplemented species were not detected? What could be the selection mechanism in BSFL guts? How do you explain that Lysinibacillus was also found to some degree in other BSFL guts that were not inoculated with the isolate?
Minor comments
- Line 26: the term „intestinal flora“ is outdated since microbes are not considered plants. Please use a different term e.g. intestinal microbiota, gut microbiome, bacterial communities…
- Line 77: were the samples cultivated/incubated only under aerobic conditions? If so, please state in the manuscript why you did not include anaerobic cultivation since the larval gut is mostly anaerobic
- Line 79: how often is “several times”?
- Line 79: How did you manage to get 1 g since larvae typically do not weight that much? Were the guts pooled? How did you sample?
- Lines 82-87: Were these media prepared from a commercially produced mix (if so, please give information from where you obtained the media) or were they mixed in your laboratory?
- Line 95: Please give the full information on the DNA extraction kit like in line 129
- Line 96: Please give more information on the PCR settings
- Line 99-100: please give more information on the sequencing procedure. What sequencing technology was used?
- Line 103: What was the humidity?
- Line 109: How often was each group replicated? I.e. how many replicates for experimental group A, B, C, …
- Line 115: Information regarding the rearing conditions should be moved to section 2.4
- Line 135: What do you mean by replicate samples? All taken from the same rearing container or from three different rearing containers? How many samples were sequenced in total?
- Line 151: Please correct “examine the structure of the gut of BSF larvae”
- Line 155: what does large-scale bacterial phyla and community complexity mean?
- Line 163: Did you normalize your data before statistical analysis? If so, how did you achieve normalization?
- Line 171: were these the only strains that you identified? I’m missing an explanation on why you chose exactly these strains
- Line 186: Please provide results on the average sequencing depth (number of sequences) after filtering. At least the average (+ standard deviation) number of sequences that were used for the subsequent bioinformatical analysis.
- Figure 2: The legend titles should be improved (“value”, “indicate value” do not say much).
- Line 209: What do you mean by “red wire frame”?
- Figure 2A and Figure 3: How do the two figures differ in terms of presented information? It is obvious that you used different types of visualization, but the information value seems to be the same.
- Line 220: The rows do not represent species. The values on the figure’s y-Axis are either taxa or genera.
- Line 224: Did you only perform pairwise comparisons using Walch’s t-test? Why not include also overall community comparison like PERMANOVA?
- Line 231, 246-247: Based on the method (16S analysis) and the type of clustering (97% OTUs) you can only say that you found the genus Lysinibacillus to be present. Based on this information it is not acceptable to state that species Lysinibacillus sphaericus is important.
- Line 244: Please check throughout the document if Genus + Species names are written in italics
- Line 254: please remove “and so on”
- Line 265: It would be more appropriate to refer to studies that were conducted with the black soldier fly instead of other insect species ([40] refers to Coleoptera, [41] refers to caterpillars, [42] refers to Isoptera). There are many studies available that investigate similar dynamics in the BSF, for example:
- https://doi.org/10.1128/AEM.01864-18
- https://doi.org/10.3389/fmicb.2021.619112
- https://doi.org/10.1007/s10482-020-01443-0
- Line 301-303: How do these results agree with the proposed BSF core microbiome? See:
- https://doi.org/10.1007/s10482-020-01443-0
- https://doi.org/10.3389/fmicb.2020.00993
- Line 308: reference is missing (Year).
- Line 314: This is a very broad assumption. There are many factors in your study that could have lead to the picture of the BSFL gut microbiome that you saw. You used sterilized substrates (see: https://doi.org/10.1093/femsec/fiab054) and you starved (see: https://www.doi.com/10.3389/fmicb.2021.601253 ) the larvae before extracting the guts. Both these factors should also be included in the discussion.
Author Response
Replies to comments on animals-1438858
In this study, the authors investigated the bacterial community in BSF gut by feeding BSF with six bacteria-inoculated foods separately. They found the increased abundance of Lysinibacillus in flies which were exposed to Lysinibacillus sphaericus. Overall, this is an interesting study, however, there are some places in the manuscript needs to be clarified.
- First, the authors should provide the rationale that why choosing these six bacteria.
Reply: Thanks for your kind suggestion. In the revision, we have added the exposition sections (See page 1, line 28-30; page 2, line 59-64).
- Second, the authors should clarify the potential applications based on this study. If inoculate with Lysinibacillus can enhance the bioconversion of waste by BSF.
Reply: Thanks for your kind suggestion. In the revision, we have added the exposition sections (See page 2, line 76-80).
- Third, clarify the reason of collecting samples at the different time points. For example, why G3 samples were collected at day8 whereas others didn’t.
Reply: Thanks very much for your kind reminder. In the revision, we have added the explaintion according to the suggestion. (See page 4, line 157-159).
- Fourth, if 24h starvation is enough for flies to remove the foreign bacteria? It would be interesting that the authors check long-term bacteria community. e.g., pupa or next generation. But this is not required in this study. A discussion would be helpful.
Reply: Thanks very much for your professional suggestion. We have added the discussion about starvation as your valuable suggestion. (See page 9, line 321-325)
- Fifth, clarify if these six bacteria used in this study are intracellular or extracellular. I don’t think the PH in flies’ gut is the major reason why Lysinibacillus could colonize the gut, since the PH is diverse in the whole intestinal tract and other bacteria can also colonize in different suitable sites.
Reply: Thanks very much for your professional suggestion. Such decomposition of microbes via gut-based mechanisms including pH, enzymes and antimicrobial proteins can explain the selective inactivation of microbes. Microbe inactivation by fly larvae depends on the specific microbe and strain. Microbes that survive the gut passage are candidates for microbes. In the revision, we have added the sentence according to the suggestion. (See page 9, line 331-338)
Minor comments:
- (in the original manuscript) - Line 23: give an explanation of functional strains;
Reply: Thanks very much for your professional and kind suggestion. We have revised according to the suggestion (See page 1, line 22-23, 28-29; page 2, line 51-55)
- (in the original manuscript) - Line 44: please give the common names if possible;
Reply: Thanks very much for the reminder. However, the common names of Trichoderma viride and Saccharomyces cerevisiae were not found. (See page 2, line 47)
- (in the original manuscript) - Line 244: should be italic.
Reply: Thanks very much for the reminder. In the revision, we have corrected it in the manuscript. (See page 8, line 285-287)

Reviewer 3 Report
In this study, the authors investigated the bacterial community in BSF gut by feeding BSF with six bacteria-inoculated foods separately. They found the increased abundance of Lysinibacillus in flies which were exposed to Lysinibacillus sphaericus.
Overall, this is an interesting study, however, there are some places in the manuscript needs to be clarified.
First, the authors should provide the rationale that why choosing these six bacteria.
Second, the authors should clarify the potential applications based on this study. If inoculate with Lysinibacillus can enhance the bioconversion of waste by BSF.
Third, clarify the reason of collecting samples at the different time points. For example, why G3 samples were collected at day8 whereas others didn’t.
Fourth, if 24h starvation is enough for flies to remove the foreign bacteria? It would be interesting that the authors check long-term bacteria community. e.g., pupa or next generation. But this is not required in this study. A discussion would be helpful.
Fifth, clarify if these six bacteria used in this study are intracellular or extracellular. I don’t think the PH in flies’ gut is the major reason why Lysinibacillus could colonize the gut, since the PH is diverse in the whole intestinal tract and other bacteria can also colonize in different suitable sites.
Minor comments:
Line 23: give an explanation of functional strains;
Line 44: please give the common names if possible;
Line 244: should be italic.
Author Response
Replies to comments on animals-1438858
In this study, the authors isolated six different strains from guts of black soldier fly larvae (BSFL) and used these strains to inoculate sterilized food wastes. Their aim was to see whether the BSFL gut microbiota can be enriched and thus shaped by adding certain functional bacterial isolates. In addition, they wanted to determine which isolated strains can establish themselves within the larval gut. The idea of improving larval digestion by adding microbes with certain beneficial features that could help to degrade hardy polysaccharides like lignin or cellulose is very interesting. However, as the authors already state in their summary, they are only presenting the results from a preliminary experiment. Thus, the spectrum of applied methods is rather narrow and does not allow for very detailed conclusions as it was attempted by the authors. Using 16S amplicon sequencing without additional quantification or detection methods is not enough to sufficiently verify the enrichment of certain species, let alone strains. With this in mind, I suggest that the conclusions should be kept more moderate and descriptive. The discussion could also be improved by referring to more fitting and topical literature on the BSF gut microbiota.
As the data comes from a preliminary study, another option would be to convert the submission into a short communication.
Response:Thank you for your comments. We carefully checked the full text as you suggested. Based on reading and summarizing literatures massively, the discussion was revised in the revision. The purpose to make a better connection with this article. The response and details are as follows.
Major comments
- Language needs to be improved throughout the manuscript. Some parts of the manuscript are not easy to understand.
Reply: Sincerely thank you for your helpful comments, and for giving us an opportunity to revise our manuscript again. In this revision, the manuscript has been carefully checked and revised again, highlighted in bule. We now hope that the paper meet the standards required.
- Line 72 (in the original manuscript) - Please add additional information on the rearing conditions of the BSF such as temperature, relative humidity, substrate that was used for rearing, container sizes, larval densities, etc.
Reply: Thank you for your comment, the detailed information on the rearing conditions of the BSF larval was on the following paragraph, “Insect rearing” (See 2.4 Insect rearing, page 3, ling 131-137)
- Material and methods should be restructured. Currently, related information is distributed across different sections.
Reply: Thank you for your comment, we have revised it in the revision. (See Materials and Methods, page 2-5, line89-203)
- Accession number (PRJNA761527) for the publicly deposited sequencing data does not work. Please make the data publicly available by re-uploading or changing the privacy settings of your project.
Reply: Thanks for your comment. The data (Accession number, PRJNA761527) has been released and available at NCBI. (See page 11, line 419)
- You did not include a “positive control” with food waste that was not sterilized. How did the sterilization of food waste affect larval development? From experience and other published studies (e.g. https://doi.org/10.1093/femsec/fiab054) we know that especially neonate larvae rely on the substrate microbiota and their development is severely inhibited if the substrates are sterilized. The missing “positive control” and the relevance of the substrate microbiota should be included in the discussion.
Reply: Thanks for your professional suggestion. The substrate has a large effect on larval gut bacterial community composition. In order to reveal the contribution of six isolated strains, respectively. To reduce the external impact, therefore, larvae fed autoclaved food waste was used the only control in this study. Thus, the positive control was not included in the revision. We have added the discussion sections in the revision (See page 9, line 313-319)
- Lines 336-338 (in the original manuscript) - Relying on 16S amplicon data alone to make such an assumption is rather ambitious. I’m missing additional (quantitative) methods to verify the gut colonization with the proposed isolates or more specifically with Lysinibacillus sphaericus.
Reply: Thanks for your professional and kind comment. The original expression was misleading, in the revision we have rephrase the sentence as you suggested. (See page 11, line 407-411)
- How do you explain that Lysinibacillus in Group B was found via 16S amplicon sequencing while the other supplemented species were not detected? What could be the selection mechanism in BSFL guts? How do you explain that Lysinibacillus was also found to some degree in other BSFL guts that were not inoculated with the isolate?
Reply: Thanks very much for your professional suggestion. Such decomposition of microbes via gut-based mechanisms including pH, enzymes and antimicrobial proteins can explain the selective inactivation of microbes. Microbe inactivation by fly larvae depends on the specific microbe and strain. Microbes that survive the gut passage are candidates for microbes. In the revision, we have added the sentence according to the suggestion. (See page 9, line 331-338)
Minor comments
- Line 26 (in the original manuscript) - the term, intestinal flora“ is outdated since microbes are not considered plants. Please use a different term e.g. intestinal microbiota, gut microbiome, bacterial communities…
Reply: Thanks very much for your professional suggestion. We have replaced “intestinal flora” by “intestinal microbiota” thoughout the text in the revision. (See page 1, line 32)
- Line 77 (in the original manuscript) - were the samples cultivated/incubated only under aerobic conditions? If so, please state in the manuscript why you did not include anaerobic cultivation since the larval gut is mostly anaerobic.
Reply: Thanks for the valuable comment. Yes, the samples cultivated only under aerobic conditions, since anaerobic bacteria would be less easy to cultivate for future industrial applications. We have stated according to the suggestion (See page 3, line 114-115)
- Line 79 (in the original manuscript) - How did you manage to get 1 g since larvae typically do not weight that much? Were the guts pooled? How did you sample?
Reply: Thanks very much for your professional comments. Yes, at least 10 larvae guts were sampled and mixed. In the revision, we have added the explanation according to the suggestion. (See page 3, line 96-100)
- Lines 82-87 (in the original manuscript) - Were these media prepared from a commercially produced mix (if so, please give information from where you obtained the media) or were they mixed in your laboratory?
Reply: Thanks for the valuable comment. These media were mixed in the laboratory. (See page 3, line 102-108)
- Line 95 (in the original manuscript) - Please give the full information on the DNA extraction kit like in line 129
Reply: Thanks very much for your reminder. In the revision, we have added the full information in parentheses according to the suggestion. (See page 3, line 121-124)
- Line 96 (in the original manuscript) - Please give more information on the PCR settings.
Reply: Thank you for your valuable comment. In the revision, we have added a detailed detailed information about PCR settings according to the suggestion. (See page 3, line 121-124)
- Line 99-100 (in the original manuscript) - please give more information on the sequencing procedure. What sequencing technology was used?
Reply: Thank you for your valuable comment. In the revision, we have added a detailed detailed information about PCR settings according to the suggestion. (See page 3, line 119-121)
- Line 103 (in the original manuscript) -What was the humidity?
Reply: Thank you for your valuable comment. We have added the detailed number of temperature (28°C) and humidity (60-70%) in the revision. (See page 3, line 132-133)
- Line 109 (in the original manuscript) - How often was each group replicated? I.e. how many replicates for experimental group A, B, C, …
Reply: Thank you for your valuable comment. We have added the detailed explanation of group replicated in the revision. (See page 4, line 151-152, 160-161)
- Line 115 (in the original manuscript) -Information regarding the rearing conditions should be moved to section 2.4
Reply: Thank you for your valuable comment. We have moved the rearing conditions to section 2.4 in the revision. (See page 3, line 134-137)
- Line 135 (in the original manuscript) - What do you mean by replicate samples? All taken from the same rearing container or from three different rearing containers? How many samples were sequenced in total?
Reply: Thank you for your comments. We have added the detailed explanation of group replicated in the revision, three larvae at each time point from same rearing container and in triplicate batches, the numbers of samples were also detailed specification. (See page 4, line 151-152, 160-161)
- Line 151 (in the original manuscript) - Please correct “examine the structure of the gut of BSF larvae”
Reply: Thanks very much for kind reminder, in the revision we have corrected them according to the suggestion. (See page 5, line 198-200)
- Line 155 (in the original manuscript) - what does large-scale bacterial phyla and community complexity mean?
Reply: Thanks very much for kind reminder, in the revision we have corrected them according to the suggestion. (See page 5, line 195-198)
- Line 163 (in the original manuscript) - Did you normalize your data before statistical analysis? If so, how did you achieve normalization?
Reply: Thanks very much for your comments. In the revision, we have added the OTU abundances were normalized by 16S rRNA copy numbers prior to the calculation of functional profiles. (See page 5, line 192-194)
- Line 171 (in the original manuscript) - were these the only strains that you identified? I’m missing an explanation on why you chose exactly these strains
Reply: Thanks very much for your comments. Yes, these are only six identified strains, about cellulase-, protease- and lipase-secreting bacteria, so we chosed. (See page 5, line 207)
- Line 186: Please provide results on the average sequencing depth (number of sequences) after filtering. At least the average (+ standard deviation) number of sequences that were used for the subsequent bioinformatical analysis.
Reply: Thanks very much for your comments. In the revision, we have added the average sequencing depth according to the suggestion. (See page 6, line 226-227)
- Figure 2: The legend titles should be improved (“value”, “indicate value” do not say much).
Reply: Thanks very much for your comments. In the revision, we have corrected them according to the suggestion. (See page 7, line247-249)
- Line 209 (in the original manuscript) - What do you mean by “red wire frame”?
Reply: Thanks very much for your comments. In the revision, we have added the red arrow according to the suggestion. (See page 7, line 249)
- Figure 2A and Figure 3: How do the two figures differ in terms of presented information? It is obvious that you used different types of visualization, but the information value seems to be the same.
Reply: Thanks very much for your comments. Figure 2 shows the average data of different sampling points, while Figure 3 is independent, and the vision and image are different for clearer explanation. So we present it separately. (See page 7, Figure2 and Figure3)
- Line 220 (in the original manuscript) -The rows do not represent species. The values on the figure’s y-Axis are either taxa or genera.
Reply: Thank you for your kind reminder, we have revised it. (See page 7, line 252-255)
- Line 224 (in the original manuscript) -Did you only perform pairwise comparisons using Walch’s t-test? Why not include also overall community comparison like PERMANOVA?
Reply: Thanks very much for your comments. Both Walch’s t-test and PERMANOVA were performed to study the significant difference. (See page 8, line 264)
- Line 231, 246-247 (in the original manuscript) - Based on the method (16S analysis) and the type of clustering (97% OTUs) you can only say that you found the genus Lysinibacillus to be present. Based on this information it is not acceptable to state that species Lysinibacillus sphaericus is important.
Reply: Thanks for your kind reminder. The original expression was misleading; we have replaced the word “important” by “particular”. (See page 8, line 273; page 9, line 289)
- Line 244 (in the original manuscript) - Please check throughout the document if Genus + Species names are written in italics
Reply: Thanks for your reminder. We have checked the entire document and corrected it in the revision. (See page 8, line 285-287)
- Line 254 (in the original manuscript) - please remove “and so on”
Reply: Thanks for your kind reminder, we have removed it in the revised manuscript. (See page 9, line 297)
- Line 265 (in the original manuscript) - It would be more appropriate to refer to studies that were conducted with the black soldier fly instead of other insect species ([40] refers to Coleoptera, [41] refers to caterpillars, [42] refers to Isoptera). There are many studies available that investigate similar dynamics in the BSF, for example:
https://doi.org/10.1128/AEM.01864-18
https://doi.org/10.3389/fmicb.2021.619112
https://doi.org/10.1007/s10482-020-01443-0
Reply: Thanks very much for the valuable suggestion, in the revision we have rephrase the references as you suggested. (See page 9, line 308-310)
- Line 301-303 (in the original manuscript) - How do these results agree with the proposed BSF core microbiome? See: https://doi.org/10.1007/s10482-020-01443-0
https://doi.org/10.3389/fmicb.2020.00993
Reply: Thanks very much for the valuable suggestion, in the revision we have rephrase the references as you suggested. (See page 10-11, line367-368, line 371-374)
- Line 308 (in the original manuscript) - reference is missing (Year).
Reply: Thanks for your reminder. We have corrected it in the revision. (See page 11, line 379)
- Line 314 (in the original manuscript) - This is a very broad assumption. There are many factors in your study that could have lead to the picture of the BSFL gut microbiome that you saw. You used sterilized substrates (see: https://doi.org/10.1093/femsec/fiab054) and you starved (see: https://www.doi.com/10.3389/fmicb.2021.601253 ) the larvae before extracting the guts. Both these factors should also be included in the discussion.
Reply: Thanks very much for your professional suggestion. We have added the discussion about sterilized substrates and starvation as your valuable suggestion. (See page 9, line 313-325)

Round 2
Reviewer 1 Report
The authors made considerable improvements in the manuscript, but there are still some points that should be addressed.
- The discussion should be revised for language and clarity. Some sentences are not well constructed.
Line 62 – From the explanation given by the authors, I would suggest to change the sentence to: “…, whereas bacterial communities in the feed residue can be affected by addition of certain bacteria”. Is this what you mean? Also in lines 184-188 the similar sentence should be reformulated to make it clearer.
Line 72 - “should be considered responsible” is very strong. Do you mean, “may be responsible”?
Line 181 – In the sentence: “However, few studies have explored the impact of isolated functional bacterial strains on intestinal microorganisms, and if they can colonised.”, do you mean: “However, few studies have explored the ability to colonize and the impact of isolated functional bacterial strains on intestinal microbiota.”?
Line 309 - “and the accession numbers are…”
Table 1 – “Genus Submission numbers” is instead “GenBank Accession numbers”? Please replace
Lines 374 – "At least 500 samples were used per treatment group"
Line 515 – The PCA results should be presented if the method is mentioned.
Line 516 – From what I understood from the results and the reply from the authors, the differences in abundances for each OTU were tested between all groups with ANOVA, followed by post-hocs tests. Is this the case? This should be detailed in the methods section. As well as the significance level, alpha, used. What post-hoc test was used? And Welch’s test was used to compare what two groups? Please detail here as well, and reformulate the result section to make it clearer (lines 754)
Line535 – Citrobacter freundii does not seem to grow better in acidic conditions – should this sentence be removed, since you added the following one as a correction to my previous comment?
Figure 3 – Thank you for adding the line plot, but the x-axis should be time. And each group should be represented with a different line. Why is the Lysinbacillus represented if the text (lines 714 onwards) only refers to Ignatzschneria across time.
Figure 5 – the y-axis is proportion, not percentage, right? Please remove % from the axis legend or convert to percentage the numbers in the axis.
Author Response
R2 - Replies to comments on animals-1438858
Reviewer 1:
The authors made considerable improvements in the manuscript, but there are still some points that should be addressed.
Response:Thank you for your comments.
- The discussion should be revised for language and clarity. Some sentences are not well constructed.
Reply: Thanks for your kind reminder. The English of this manuscript has been revised by a native English-speaker engaged through the auspices of a professional proofreading service. Language Revision Certificate is shown in second page of this response letter.
- Line 62 – From the explanation given by the authors, I would suggest to change the sentence to: “…, whereas bacterial communities in the feed residue can be affected by addition of certain bacteria”. Is this what you mean? Also in lines 184-188 the similar sentence should be reformulated to make it clearer.
Reply: Thanks very much for your comment. Yes, your understanding is accurate. In the revision, we have added and modified the sentence as you suggested. (See page 2, line 65-66, 90-91)
- Line 72 - “should be considered responsible” is very strong. Do you mean, “may be responsible”?
Reply: Thanks for your kind reminder. Yes, your understanding is accurate. In the revision, we have modified the sentence as you suggested. (See page 1, line 30)
- Line 181 – In the sentence: “However, few studies have explored the impact of isolated functional bacterial strains on intestinal microorganisms, and if they can colonised.”, do you mean: “However, few studies have explored the ability to colonize and the impact of isolated functional bacterial strains on intestinal microbiota.”?
Reply: Thanks very much for your comment. Yes, your understanding is accurate. In the revision, we have modified the sentence as you suggested. (See page 2, line 82-83)
- Line 309 - “and the accession numbers are…”
Reply: Thanks very much for your professional suggestion. We have revised it in the revision. (See page 3, line 132)
- Table 1 – “Genus Submission numbers” is instead “GenBank Accession numbers”? Please replace
Reply: Thanks for your kind reminder. We have replaced “Genus Submission numbers” by “GenBank Accession numbers” in Table 1. (See page 4, line 150)
- Lines 374 – "At least 500 samples were used per treatment group".
Reply: Thanks very much for your comment. We have added the words it in the revision. (See page 4, line 152)
- Line 515 – The PCA results should be presented if the method is mentioned.
Reply: Thank you for your comments. Since the results have not been explained, so we deleted this sentence. (See page 5, line 198)
- Line 516 – From what I understood from the results and the reply from the authors, the differences in abundances for each OTU were tested between all groups with ANOVA, followed by post-hocs tests. Is this the case? This should be detailed in the methods section. As well as the significance level, alpha, used. What post-hoc test was used? And Welch’s test was used to compare what two groups? Please detail here as well, and reformulate the result section to make it clearer (lines 754)
Reply: Thanks very much for your professional suggestion. Yes, the differences in abundances for each OTU were tested between all groups with ANOVA, followed by post-hocs tests. In the revision, we have added the sentence “The intergroup comparisons were performed by using a one-way analysis of variance (ANOVA) followed by Scheffe's post hoc test (P < 0.05).” (See page 5, line 198-200).
The results of welch’s t-test were not shown, so we deleted the sentence “Differences in bacterial abundance between groups were analysed by Welch’s t-test of Statistical Analyses of Metagenomic Profiles STAMP.”
- Line 535 – Citrobacter freundii does not seem to grow better in acidic conditions – should this sentence be removed, since you added the following one as a correction to my previous comment?
Reply: Thank you for your comments. In the revision, we have removed the sentence as you suggested. And the following one is as a correction to your previous comment. (See page 4, line 216)
Figure 3 – Thank you for adding the line plot, but the x-axis should be time. And each group should be represented with a different line. Why is the Lysinbacillus represented if the text (lines 714 onwards) only refers to Ignatzschneria across time.
Reply: Thank you for your comments. The line plot in the figure 3, the x-axis are the samples, and the y-axis are the relative abundance. At genus-level, Ignatzschineria was the least abundant among the seven groups on the first day of sampling. (See page 7, figure 3)
Figure 5 – the y-axis is proportion, not percentage, right? Please remove % from the axis legend or convert to percentage the numbers in the axis.
Reply: Thanks for your kind reminder. Yes, you are right. In the revision, we have removed the “%” in Figure 5. (See page 10, figure 5)

Reviewer 2 Report
I appreciate the efforts of the authors to address my concerns and implement the suggested revisions, however, the submitted revision was done in a rather superficial way and the manuscript still needs some work. Many of the added text passages are hard to understand or do not make much sense.
1. Language needs to be improved throughout the manuscript. Some parts of the manuscript are not easy to understand.
Author Reply: Sincerely thank you for your helpful comments, and for giving us an opportunity to revise our manuscript again. In this revision, the manuscript has been carefully checked and revised again, highlighted in bule. We now hope that the paper meet the standards required.
REPLY: Unfortunately, language still needs thorough revision. The authos should consider to making use of English editing services.
2. Line 72 (in the original manuscript) - Please add additional information on the rearing conditions of the BSF such as temperature, relative humidity, substrate that was used for rearing, container sizes, larval densities, etc.
Author Reply: Thank you for your comment, the detailed information on the rearing conditions of the BSF larval was on the following paragraph, “Insect rearing” (See 2.4 Insect rearing, page 3, ling 131-137)
REPLY: Thank you for providing the additional information. It would be great if you could add an approximate number of larvae that hatched from the 2 colonized each box. The larval density is a highly relevant parameter for multiple dynamics taking place especially in terms of microbiome shifts and the amount of eggs alone does not help much to compare studies.
4. Accession number (PRJNA761527) for the publicly deposited sequencing data does not work. Please make the data publicly available by re-uploading g and that subsequently or changing the privacy settings of your project.
Author Reply: Thanks for your comment. The data (Accession number, PRJNA761527) has been released and available at NCBI. (See page 11, line 419)
REPLY: Thank you for releasing the data. In your manuscript, you state that you used the Illumina MiSeq platform (Line 186-187) to generate the sequence data. The information in your PRJNA761527 BioProject, however, says that you used the IonTorrent S5 XL platform. Which information is correct?
7. How do you explain that Lysinibacillus in Group B was found via 16S amplicon sequencing while the other supplemented species were not detected? What could be the selection mechanism in BSFL guts? How do you explain that Lysinibacillus was also found to some degree in other BSFL guts that were not inoculated with the isolate?
Author Reply: Thanks very much for your professional suggestion. Such decomposition of microbes via gut-based mechanisms including pH, enzymes and antimicrobial proteins can explain the selective inactivation of microbes. Microbe inactivation by fly larvae depends on the specific microbe and strain. Microbes that survive the gut passage are candidates for microbes. In the revision, we have added the sentence according to the suggestion. (See page 9, line 331-338)
REPLY: This is a rather general answer. A great part of the text that was added in lines 331-338 does not make sense or is not clear, for example:
- What do you mean by “Escherichia coli and Bacillus subtilis through the gut passage was identified completely inactivation by using the fluorescent bacteria, and Enterococcus faecalis for BSF larvae had low reductions in the larvae and residue.” (Line 335-337)
- Or “Microbes that survive the gut passage are candidates for microbes.” (line 338-339)
9. Line 77 (in the original manuscript) - were the samples cultivated/incubated only under aerobic conditions? If so, please state in the manuscript why you did not include anaerobic cultivation since the larval gut is mostly anaerobic.
Author Reply: Thanks for the valuable comment. Yes, the samples cultivated only under aerobic conditions, since anaerobic bacteria would be less easy to cultivate for future industrial applications. We have stated according to the suggestion (See page 3, line 114-115)
REPLY: Thank you for adding some explanation. However, I think that excluding a highly relevant method for this context just because it is “not easy” is not a valid excuse. At this point, it is too late to add it to the manuscript, but the authors should consider this for future experiments.
10. Line 79 (in the original manuscript) - How did you manage to get 1 g since larvae typically do not weight that much? Were the guts pooled? How did you sample?
Author Reply: Thanks very much for your professional comments. Yes, at least 10 larvae guts were sampled and mixed. In the revision, we have added the explanation according to the suggestion. (See page 3, line 96-100)
REPLY: Thank you for adding the additional information. I’m still surprised that you manage to obtain 1 g of guts with only ten larvae that on average weight 200 mg as a whole at their biomass peak
16. Line 109 (in the original manuscript) - How often was each group replicated? I.e. how many replicates for experimental group A, B, C, …
Author Reply: Thank you for your valuable comment. We have added the detailed explanation of group replicated in the revision. (See page 4, line 151-152, 160-161)
REPLY: It is not clear to me what you mean by “At least 500 samples per treatment group” (Line 150) or what this number refers to. Please rephrase or explain how you calculate this number and what kind of samples you mean. It would be sufficient to state for example “Guts from X larvae per sample were pooled for DNA extraction, resulting in a total of X biological replicates per time point for each treatment. A total of X samples were subsequently submitted for sequencing.”
21. Line 163 (in the original manuscript) - Did you normalize your data before statistical analysis? If so, how did you achieve normalization?
Author Reply: Thanks very much for your comments. In the revision, we have added the OTU abundances were normalized by 16S rRNA copy numbers prior to the calculation of functional profiles. (See page 5, line 192-194)
REPLY: Thank you for providing additional information, however, this does not make a lot of sense to me. You did not show any results on functional profiles that you are mentioning? By normalization I was referring to equalizing sequencing depth. Please see: https://doi.org/10.1186/s40168-017-0237-y
24. Figure 2: The legend titles should be improved (“value”, “indicate value” do not say much).
Author Reply: Thanks very much for your comments. In the revision, we have corrected them according to the suggestion. (See page 7, line247-249)
REPLY: Thank you for addressing this issue, however, I was referring to the legend titles WITHIN the figure (on top of the color gradient and bubble sizes)
28. Line 224 (in the original manuscript) -Did you only perform pairwise comparisons using Walch’s t-test? Why not include also overall community comparison like PERMANOVA?
Author Reply: Thanks very much for your comments. Both Walch’s t-test and PERMANOVA were performed to study the significant difference. (See page 8, line 264)
REPLY: In line 264 you state that you applied ANOVA. This is not the same as PERMANOVA. No information on this statistical analysis (ANOVA or PERMANOVA) was given in the material and methods section.
35. Line 314 (in the original manuscript) - This is a very broad assumption. There are many factors in your study that could have lead to the picture of the BSFL gut microbiome that you saw. You used sterilized substrates (see: https://doi.org/10.1093/femsec/fiab054) and you starved (see: https://www.doi.com/10.3389/fmicb.2021.601253 ) the larvae before extracting the guts. Both these factors should also be included in the discussion.
Author reply: Thanks very much for your professional suggestion. We have added the discussion about sterilized substrates and starvation as your valuable suggestion. (See page 9, line 313-325)
REPLY: I have the impression that the revision was done in a rather superficial way as there have been added many sentences throughout the manuscript that do not make sense, e.g. also in this section:
- Line 320 “In order to reveal the contribution of six isolated strains, respectively.”
- Line 323-326 “However, in order to reduce the impact of food residues in the intestine on intestinal microorganisms, according recent study [58], the collected larvae were starved for 24 h to empty their ingested contents before DNA extraction.”
Author Response
R2 - Replies to comments on animals-1438858
Reviewer 2:
I appreciate the efforts of the authors to address my concerns and implement the suggested revisions, however, the submitted revision was done in a rather superficial way and the manuscript still needs some work. Many of the added text passages are hard to understand or do not make much sense.
Response:Thank you for your comments.
- Language needs to be improved throughout the manuscript. Some parts of the manuscript are not easy to understand.
Author Reply: Sincerely thank you for your helpful comments, and for giving us an opportunity to revise our manuscript again. In this revision, the manuscript has been carefully checked and revised again, highlighted in bule. We now hope that the paper meet the standards required.
REPLY: Unfortunately, language still needs thorough revision. The authos should consider to making use of English editing services.
R2-Reply: Thanks for your kind reminder. The English of this manuscript has been revised by a native English-speaker engaged through the auspices of a professional proofreading service. Language Revision Certificate is shown in second page of this response letter.
- Line 72 (in the original manuscript) - Please add additional information on the rearing conditions of the BSF such as temperature, relative humidity, substrate that was used for rearing, container sizes, larval densities, etc.
Author Reply: Thank you for your comment, the detailed information on the rearing conditions of the BSF larval was on the following paragraph, “Insect rearing” (See 2.4 Insect rearing, page 3, ling 131-137)
REPLY: Thank you for providing the additional information. It would be great if you could add an approximate number of larvae that hatched from the 2 colonized each box. The larval density is a highly relevant parameter for multiple dynamics taking place especially in terms of microbiome shifts and the amount of eggs alone does not help much to compare studies.
R2-Reply: Thanks very much for the valuable suggestion. However, since the number of samples used in the experiment is small, it is with great regret that the density of insects is not counted. And we will add this index in subsequent experiments. (See page 3, line 134)
- Accession number (PRJNA761527) for the publicly deposited sequencing data does not work. Please make the data publicly available by re-uploading g and that subsequently or changing the privacy settings of your project.
Author Reply: Thanks for your comment. The data (Accession number, PRJNA761527) has been released and available at NCBI. (See page 11, line 419)
REPLY: Thank you for releasing the data. In your manuscript, you state that you used the Illumina MiSeq platform (Line 186-187) to generate the sequence data. The information in your PRJNA761527 BioProject, however, says that you used the IonTorrent S5 XL platform. Which information is correct?
R2-Reply: Thanks for your kind reminder. The expression of IonTorrent S5 XL platform was misleading, we have replaced “IonTorrent S5 XL platform” by “Illumina MiSeq platform” on NCBI. In this study, 16S rRNA sequencing was sequenced was on an Illumina MiSeq platform (Illumina, San Diego, CA, USA) in PE300 mode. (See page 5, line 186-187)
- How do you explain that Lysinibacillus in Group B was found via 16S amplicon sequencing while the other supplemented species were not detected? What could be the selection mechanism in BSFL guts? How do you explain that Lysinibacillus was also found to some degree in other BSFL guts that were not inoculated with the isolate?
Author Reply: Thanks very much for your professional suggestion. Such decomposition of microbes via gut-based mechanisms including pH, enzymes and antimicrobial proteins can explain the selective inactivation of microbes. Microbe inactivation by fly larvae depends on the specific microbe and strain. Microbes that survive the gut passage are candidates for microbes. In the revision, we have added the sentence according to the suggestion. (See page 9, line 331-338)
REPLY: This is a rather general answer. A great part of the text that was added in lines 331-338 does not make sense or is not clear, for example:
- What do you mean by “Escherichia coli and Bacillus subtilis through the gut passage was identified completely inactivation by using the fluorescent bacteria, and Enterococcus faecalis for BSF larvae had low reductions in the larvae and residue.” (Line 335-337)
- Or “Microbes that survive the gut passage are candidates for microbes.” (line 338-339)
R2-Reply: Thanks very much for your comment. To explain that Lysinibacillus in Group B was found while the other supplemented species were not detected. The detailed explanation as follows: Bacteria serve directly as food for fly larvae and help decompose macronutrients. (See page 9, Line 330) So the next sentence, “Escherichia coli and Bacillus subtilis through the gut passage was identified completely inactivation by using the fluorescent bacteria, and Enterococcus faecalis for BSF larvae had low reductions in the larvae and residue.” (See page 9, Line 333-335) The last sentence, “Microbes that survive the gut passage are candidates for microbes” (See page 9, Line 336-337)
- Line 77 (in the original manuscript) - were the samples cultivated/incubated only under aerobic conditions? If so, please state in the manuscript why you did not include anaerobic cultivation since the larval gut is mostly anaerobic.
Author Reply: Thanks for the valuable comment. Yes, the samples cultivated only under aerobic conditions, since anaerobic bacteria would be less easy to cultivate for future industrial applications. We have stated according to the suggestion (See page 3, line 114-115)
REPLY: Thank you for adding some explanation. However, I think that excluding a highly relevant method for this context just because it is “not easy” is not a valid excuse. At this point, it is too late to add it to the manuscript, but the authors should consider this for future experiments.
R2-Reply: Thanks very much for your professional suggestion, we very much appreciate your understanding and support. In fact, we have also noticed this defect. In the next design experiments, the anaerobic bacteria of BSF larvae will be studied by using both culture-independent and dependent methods.
- Line 79 (in the original manuscript) - How did you manage to get 1 g since larvae typically do not weight that much? Were the guts pooled? How did you sample?
Author Reply: Thanks very much for your professional comments. Yes, at least 10 larvae guts were sampled and mixed. In the revision, we have added the explanation according to the suggestion. (See page 3, line 96-100)
REPLY: Thank you for adding the additional information. I’m still surprised that you manage to obtain 1 g of guts with only ten larvae that on average weight 200 mg as a whole at their biomass peak
R2-Reply: Thanks very much for your kind reminder. At least 10 larvae guts were sampled and mixed, the expression of 1g was misleading. After verification, we have replaced “1 g” by “0.1 g” in the revision. (See page 3, line 103)
- Line 109 (in the original manuscript) - How often was each group replicated? I.e. how many replicates for experimental group A, B, C, …
Author Reply: Thank you for your valuable comment. We have added the detailed explanation of group replicated in the revision. (See page 4, line 151-152, 160-161)
REPLY: It is not clear to me what you mean by “At least 500 samples per treatment group” (Line 150) or what this number refers to. Please rephrase or explain how you calculate this number and what kind of samples you mean. It would be sufficient to state for example “Guts from X larvae per sample were pooled for DNA extraction, resulting in a total of X biological replicates per time point for each treatment. A total of X samples were subsequently submitted for sequencing.”
R2-Reply: Thanks for your comment. At least 500 samples were reared per treatment group, the subsequent test samples were selected from them. So we highlighted this number. In the revision, we have revised the sentence: “Guts from 3 larvae per sample were pooled for DNA extraction, resulting in a total of 3 biological replicates per time point for each treatment. A total of 47 samples were subsequently submitted for sequencing.” (See page 4, line 161-164)
- Line 163 (in the original manuscript) - Did you normalize your data before statistical analysis? If so, how did you achieve normalization?
Author Reply: Thanks very much for your comments. In the revision, we have added the OTU abundances were normalized by 16S rRNA copy numbers prior to the calculation of functional profiles. (See page 5, line 192-194)
REPLY: Thank you for providing additional information, however, this does not make a lot of sense to me. You did not show any results on functional profiles that you are mentioning? By normalization I was referring to equalizing sequencing depth. Please see: https://doi.org/10.1186/s40168-017-0237-y
R2-Reply: Thanks very much for your professional suggestion. We misunderstood your meaning. To truthfully reflect the authenticity of the data, it was not normalize before statistical analysis, we have deleted the added sentence. (See page 5, line 197)
- Figure 2: The legend titles should be improved (“value”, “indicate value” do not say much).
Author Reply: Thanks very much for your comments. In the revision, we have corrected them according to the suggestion. (See page 7, line247-249)
REPLY: Thank you for addressing this issue, however, I was referring to the legend titles WITHIN the figure (on top of the color gradient and bubble sizes)
R2-Reply: Thanks for your kind reminder. In the description of figure 2, in the value label, the yellower and larger the bubble, the higher the species abundance. (See page 7, line247-248)
- Line 224 (in the original manuscript) -Did you only perform pairwise comparisons using Walch’s t-test? Why not include also overall community comparison like PERMANOVA?
Author Reply: Thanks very much for your comments. Both Walch’s t-test and PERMANOVA were performed to study the significant difference. (See page 8, line 264)
REPLY: In line 264 you state that you applied ANOVA. This is not the same as PERMANOVA. No information on this statistical analysis (ANOVA or PERMANOVA) was given in the material and methods section.
R2-Reply: Thanks very much for your professional suggestion. In the revision, we have added the sentence in 2.7. Bioinformatics and Statistical analyses. (See page 5, line 198-200)
- Line 314 (in the original manuscript) - This is a very broad assumption. There are many factors in your study that could have lead to the picture of the BSFL gut microbiome that you saw. You used sterilized substrates (see: https://doi.org/10.1093/femsec/fiab054) and you starved (see: https://www.doi.com/10.3389/fmicb.2021.601253 ) the larvae before extracting the guts. Both these factors should also be included in the discussion.
Author reply: Thanks very much for your professional suggestion. We have added the discussion about sterilized substrates and starvation as your valuable suggestion. (See page 9, line 313-325)
REPLY: I have the impression that the revision was done in a rather superficial way as there have been added many sentences throughout the manuscript that do not make sense, e.g. also in this section:
- Line 320 “In order to reveal the contribution of six isolated strains, respectively.”
- Line 323-326 “However, in order to reduce the impact of food residues in the intestine on intestinal microorganisms, according recent study [58], the collected larvae were starved for 24 h to empty their ingested contents before DNA extraction.”
R2-Reply: Thanks very much for your comments. In order to maintain the desired sentence flow, we have deleated the line 320 “In order to reveal the contribution of six isolated strains, respectively.” and “However” . And we have rephrase the sentence as you suggested in the revision. (See page 9, line 321-324)

Round 3
Reviewer 2 Report
The use of professional proofreading services greatly improved the submission. The authors were eager to address my and the other reviewer's criticisms and suggestions for improvement, and were able to significantly improve the manuscript over the past review rounds.
I remain with a few minor suggestions:
Line 29-31: How can the six strains "be considered responsible for the functional characteristics of larvae" when Lysinibacillus is the only one that accumulates in the gut?
Line 76: "efficiency of conversion" (“of” is missing)
Line 170-172: In line 161-163 you state that you used three larvae per sample, but here you write about 50-200 larvae that were used for DNA extraction. Could you please clarify what you mean by 50-200 larvae?
Line 172-173: What do you mean by “three independent batches”? Are these the three biological replicates you mention in lines 162-163? Please clarify in the text otherwise it is a bit confusing.
Line 214: space is missing between “with” and “a”
Line 226: Phylum names do not need to be written in italics, only genus names (please also check table 1 and throughout the document)
Line 302: There seems to be a small mistake in the sentence. I suggest to change it to “…mosquitoes and bioremediation [44,45], which as was initially isolated from an adult black fly in Nigeria [46].”
Line 299: Repetition: please delete one of the excess “lignin-degrading bacteria”
Line 346-348: Unfortunately, with the methods you applied you cannot say if aerobic bacteria are actively growing in the gut. By extracting DNA and sequencing it you can only tell that the DNA of a specific organism was present in the larval gut at some point, but you cannot say whether it was alive or even growing/reproducing. It could also be that Lysinibacillus was taken up in significantly higher amounts than the other six bacteria and the cells died in the guts.
Line 407: “inoculation method” instead of “inoculated method”
Author Response
R3 - Replies to comments on animals-1438858
The use of professional proofreading services greatly improved the submission. The authors were eager to address my and the other reviewer's criticisms and suggestions for improvement, and were able to significantly improve the manuscript over the past review rounds. I remain with a few minor suggestions:
Response:Thank you for your comments.
- Line 29-31: How can the six strains "be considered responsible for the functional characteristics of larvae" when Lysinibacillus is the only one that accumulates in the gut? These six strains were isolated and identified from selective medium
Reply: Thanks for your professional and kind comment. The original expression was misleading; In the revision, we have rephrased this reference “These six strains were isolated on selective medium from larval intestinal track, and identified and screened,” (See page 1, line 30)
- Line 76: "efficiency of conversion" (“of” is missing)
Reply: Thanks for your kind reminder. In the revision we have corrected. (See page 2, line 76)
- Line 170-172: In line 161-163 you state that you used three larvae per sample, but here you write about 50-200 larvae that were used for DNA extraction. Could you please clarify what you mean by 50-200 larvae?
Reply: Thanks for your professional and kind reminder. The original expression was misleading; we have rephrased the references “For each batch, a total of 141 larvae from 47 sampling points were rinsed with sterile water after alcohol cleaning, and anatomized to extract total DNA from the gut.”(See page 4, line 170-172)
- Line 172-173: What do you mean by “three independent batches”? Are these the three biological replicates you mention in lines 162-163? Please clarify in the text otherwise it is a bit confusing.
Reply: Thanks for your professional and kind reminder. Yes, they are the three biological replicates that mention in lines 162-163. The original expression was misleading; we have rephrased the references the same as above. (See page 4, line 170-172)
- Line 214: space is missing between “with” and “a”
Reply: Thanks for your kind reminder. In the revision we have revised as you suggested. (See page 5, line 214)
- Line 226: Phylum names do not need to be written in italics, only genus names (please also check table 1 and throughout the document)
Reply: Thanks for your kind reminder. We have checked the entire test and corrected in the revision. (See page 6, line 226)
- Line 302: There seems to be a small mistake in the sentence. I suggest to change it to “…mosquitoes and bioremediation [44,45], which aswas initially isolated from an adult black fly in Nigeria [46].”
Reply: Thanks for your kind reminder. In the revision we have revised as you suggested. (See page 9, line 291)
- Line 299: Repetition: please delete one of the excess “lignin-degrading bacteria”
Reply: Thanks for your kind reminder. In the revision we have corrected as you suggested. (See page 9, line 299)
- Line 346-348: Unfortunately, with the methods you applied you cannot say if aerobic bacteria are actively growing in the gut. By extracting DNA and sequencing it you can only tell that the DNA of a specific organism was present in the larval gut at some point, but you cannot say whether it was alive or even growing/reproducing. It could also be that Lysinibacilluswas taken up in significantly higher amounts than the other six bacteria and the cells died in the guts.
Reply: Thanks for your kind reminder. In the revision we have deleted the sentence according your suggested. (See page 10, line 346)
- Line 407: “inoculation method” instead of “inoculated method”
Reply: Thanks for your suggestion, we have replaced “inoculated method” by “inoculation method”. (See page 11, line 405)
